# Learning to Predict Trustworthiness with Steep Slope Loss

**Yan Luo**[†], **Yongkang Wong**[‡], **Mohan S Kankanhalli**[‡], **Qi Zhao**[†]

[†] Department of Computer Science & Engineering, University of Minnesota
[‡] School of Computing, National University of Singapore

luoxx648@umn.edu, yongkang.wong@nus.edu.sg, mohan@comp.nus.edu.sg, qzhao@cs.umn.edu

## Abstract

Understanding the trustworthiness of a prediction yielded by a classifier is critical for the safe and effective use of AI models. Prior efforts have been proven to be reliable on small-scale datasets. In this work, we study the problem of predicting trustworthiness on real-world large-scale datasets, where the task is more challenging due to high-dimensional features, diverse visual concepts, and a large number of samples. In such a setting, we observe that the trustworthiness predictors trained with prior-art loss functions, i.e., the cross entropy loss, focal loss, and true class probability confidence loss, are prone to view both correct predictions and incorrect predictions to be trustworthy. The reasons are two-fold. Firstly, correct predictions are generally dominant over incorrect predictions. Secondly, due to the data complexity, it is challenging to differentiate the incorrect predictions from the correct ones on real-world large-scale datasets. To improve the generalizability of trustworthiness predictors, we propose a novel *steep slope loss* to separate the features w.r.t. correct predictions from the ones w.r.t. incorrect predictions by two slide-like curves that oppose each other. The proposed loss is evaluated with two representative deep learning models, i.e., Vision Transformer and ResNet, as trustworthiness predictors. We conduct comprehensive experiments and analyses on ImageNet, which show that the proposed loss effectively improves the generalizability of trustworthiness predictors. The code and pre-trained trustworthiness predictors for reproducibility are available at https://github.com/luoyan407/predict_trustworthiness.

## 1 Introduction

Classification is a ubiquitous learning problem that categorizes objects according to input features. It is widely used in a range of applications, such as robotics [1], environment exploration [2], medical diagnosis [3], etc. In spite of the successful development of deep learning methods in recent decades, high-performance classifiers would still have a chance to make mistakes due to the improvability of models and the complexity of real-world data [4, 5, 6, 7].

To assess whether the prediction yielded by a classifier can be trusted or not, there are growing efforts towards learning to predict trustworthiness [8, 9]. These methods are evaluated on small-scale datasets, e.g., MNIST [10], where the data is relatively simple and existing classifiers have achieved high accuracy ($> 99\%$). As a result, there are a dominant proportion of correct predictions and the trustworthiness predictors are prone to classify incorrect predictions as trustworthy predictions. The characteristics that the simple data is easy-to-classify aggravate the situation. To further understand the prowess of predicting trustworthiness, we study this problem on the real-world large-scale datasets, i.e., ImageNet [11]. This is a challenging theme for classification in terms of boundary complexity, class ambiguity, and feature dimensionality [12]. As a result, failed predictions are inevitable.

35th Conference on Neural Information Processing Systems (NeurIPS 2021).

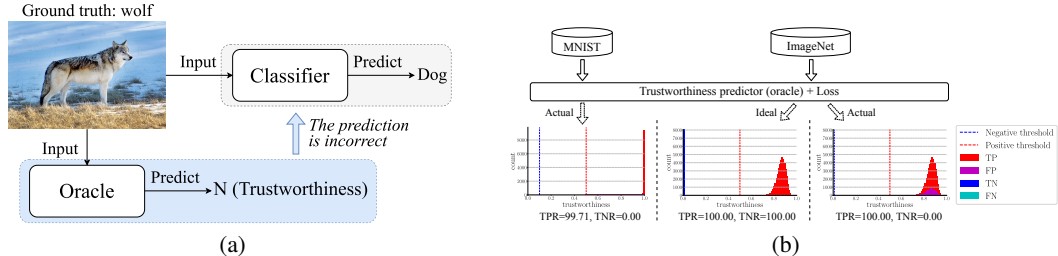

Figure 1: Conceptual illustrations of trustworthiness prediction. (a) shows the process of predicting trustworthiness where the oracle is the trustworthiness predictor. The illustration in (b) shows that the task is challenging on ImageNet, where TCP's confidence loss [9] is used in this example. The confidence that is greater (lower) than the positive (negative) threshold would be classified as a trustworthy (untrustworthy) prediction. Usually, both the positive threshold and the negative threshold are 0.5, but the negative threshold is $\frac{1}{\text{\# of classes}}$ in the case of TCP.

A general illustration of predicting trustworthiness [9, 13] is shown in Fig. 1a. The trustworthiness predictor [13] that is based on the maximum confidence have been proven to be unreliable [8, 14, 15, 16]. Instead, Corbiere et al. [9] propose the true class probability (TCP) that uses the confidence w.r.t. the ground-truth class to determine whether to trust the classifier's prediction or not. Nevertheless, the classification confidence is sensitive to the data. As shown in Fig. 1b, TCP predicts that all the incorrect predictions (0.9% in predictions) are trustworthy on MNIST [10] and predicts that all the incorrect predictions (∼16% in predictions) are trustworthy on ImageNet [11].

To comprehensively understand this problem, we follow the learning scheme used in [9] and use two state-of-the-art backbones, i.e., ViT [7] and ResNet [5], as the trustworthiness predictors. For simplicity, we call the "trustworthiness predictor" an *oracle*. We find that the oracles trained with cross entropy loss [17], focal loss [18], and TCP confidence loss [9] on ImageNet are prone to overfit the training samples, i.e., the true positive rate (TPR) is close to 100% while the true negative rate (TNR) is close to 0%. To improve the generalizability of oracles, we propose a novel loss function named the steep slope loss. The proposed steep slope loss consists of two slide-like curves that cross with each other and face in the opposite direction to separate the features w.r.t. trustworthy and untrustworthy predictions. It is tractable to control the slopes by indicating the heights of slides. In this way, the proposed loss is able to be flexible and effective to push the features w.r.t. correct and incorrect predictions to the well-classified regions.

Predicting trustworthiness is similar to as well as different from conventional classification tasks. On one hand, predicting trustworthiness can be formulated as a binary classification problem. On the other hand, task-specific semantics are different between the classification task and predicting trustworthiness. The classes are referred to visual concepts, such as dog, cat, etc., in the classification task, while the ones in predicting trustworthiness are abstract concepts. The trustworthiness could work on top of the classes in the classification task. In other words, the classes in the classification task are specific and closed-form, while trustworthiness is open-form and is related to the classes in the classification task.

The contribution of this work can be summarized as follows.

- We study the problem of predicting trustworthiness with widely-used classifiers on ImageNet. Specifically, we observe that a major challenge of this learning task is that the cross entropy loss, focal loss, and TCP loss are prone to overfit the training samples, where correct predictions are dominant over incorrect predictions.

- We propose the steep slope loss function that improves the generalizability of trustworthiness predictors. We conduct comprehensive experiments and analyses, such as performance on both small-scale and large-scale datasets, analysis of distributions separability, comparison to the class-balanced loss, etc., which verify the efficacy of the proposed loss.

- To further explore the practicality of the proposed loss, we train the oracle on the ImageNet training set and evaluate it on two variants of ImageNet validation set, i.e., the stylized validation set and the adversarial validation set. The two variants' domains are quite different

from the domain of the training set. We find that the learned oracle is able to consistently differentiate the trustworthy predictions from the untrustworthy predictions.

## 2 Preliminaries

In this section, we first recap how a deep learning model learns in the image classification task. Then, we show how the task of predicting trustworthiness connects to the classification task.

**Supervised Learning for Classification**. In classification tasks, given a training sample, i.e., image $\boldsymbol{x} \in \mathbb{R}^m$ and corresponding ground-truth label $y \in \mathcal{Y} = \{1, \ldots, K\}$, we assume that samples are drawn i.i.d. from an underlying distribution. The goal of the learning task is to learn to find a classifier $f^{(cls)}(\cdot; \theta')$ with training samples for classification. $\theta'$ is the set of parameters of the classifier. Let $f_{\theta'}^{(cls)}(\cdot) = f^{(cls)}(\cdot; \theta')$. The optimization problem is defined as

$$f_{\theta'}^{*(cls)} = \underset{f_{\theta'}^{(cls)}}{\arg \min} \hat{\mathcal{R}}(f_{\theta'}^{(cls)}, \ell^{(cls)}, D_{tr}), \tag{1}$$

where $f_{\theta'}^{*(cls)}$ is the learned classifier, $\hat{\mathcal{R}}$ is the empirical risk, $\ell^{(cls)}$ is a loss function for classification, and $D_{tr}$ is the set of training samples.

**Supervised Learning for Predicting Trustworthiness**. In contrast to the learning task for classification, which is usually a multi-class single-label classification task [4, 5, 6, 7], learning to predict trustworthiness is a binary classification problem, where the two classes are positive (i.e., trustworthy) or negative (i.e., untrustworthy). Similar to [9], given a pair $(\boldsymbol{x}, y)$ and a classifier $f_{\theta'}^{(cls)}$, we define the ground-truth label $o$ for predicting trustworthiness as

$$o = \begin{cases} 1, & \text{if } \arg \max f_{\theta'}^{(cls)}(\boldsymbol{x}) = y \\ 0, & \text{otherwise} \end{cases} \tag{2}$$

In other words, the classifier correctly predicts the image's label so the prediction is trustworthy in hindsight, otherwise the prediction is untrustworthy.

The learning task for predicting trustworthiness follows a similar learning framework in the classification task. Let $f_\theta(\cdot)$ be an oracle (i.e., a trustworthiness predictor). A generic loss function $\ell : \mathbb{R}^m \times \mathbb{R} \to \mathbb{R}_{\geq 0}$, where $\mathbb{R}_{\geq 0}$ is a non-negative space and $m$ is the number of classes. Given training samples $(\boldsymbol{x}, y) \in D_{tr}$, the optimization problem for predicting trustworthiness is defined as

$$f_\theta^* = \underset{f_\theta}{\arg \min} \frac{1}{|D_{tr}|} \sum_{i=1}^{|D_{tr}|} \ell(f_\theta(\boldsymbol{x}_i), o_i), \tag{3}$$

where $|D_{tr}|$ is the cardinality of $D_{tr}$.

Particularly, we consider two widely-used loss functions for classification and the loss function used for training trustworthiness predictors as baselines. They are the cross entropy loss [17], focal loss [18], and TCP confidence loss [9]. Let $p(o = 1|\theta, \boldsymbol{x}) = 1/(1 + \exp(-z))$ be the trustworthiness confidence, where $z \in \mathbb{R}$ is the descriminative feature produced by the oracle, i.e., $z = f_\theta(\boldsymbol{x})$. The three loss functions can be written as

$$\ell_{CE}(f_\theta(\boldsymbol{x}), o) = -o \cdot \log p(o = 1|\theta, \boldsymbol{x}) - (1 - o) \cdot \log(1 - p(o = 1|\theta, \boldsymbol{x})), \tag{4}$$

$$\ell_{Focal}(f_\theta(\boldsymbol{x}), o) = -o \cdot (1 - p(o = 1|\theta, \boldsymbol{x}))^\gamma \log p(o = 1|\theta, \boldsymbol{x}) - \\ (1 - o) \cdot (p(o = 1|\theta, \boldsymbol{x}))^\gamma \log(1 - p(o = 1|\theta, \boldsymbol{x})), \tag{5}$$

$$\ell_{TCP}(f_\theta(\boldsymbol{x}), y) = (f_\theta(\boldsymbol{x}) - p(\hat{y} = y|\theta', \boldsymbol{x}))^2. \tag{6}$$

In the focal loss, $\gamma$ is a hyperparameter. In the TCP confidence loss, $\hat{y}$ is the predicted label and $p(\hat{y} = y|\theta', \boldsymbol{x})$ is the classification probability w.r.t. the ground-truth class.

Consequently, the learned oracle would yield $z$ to generate the trustworthiness confidence. In the cases of $\ell_{CE}$ and $\ell_{Focal}$, the oracle considers a prediction is trustworthy if the corresponding trustworthiness confidence is greater than the positive threshold 0.5, i.e., $p(o = 1|\theta, \boldsymbol{x}) > 0.5$. The predictions whose trustworthiness confidences are equal to or lower than the negative threshold 0.5 are viewed to be untrustworthy. In the case of $\ell_{TCP}$, the positive threshold is also 0.5, but the negative threshold correlates to the number of classes in the classification task. It is defined as $1/K$ in [9].

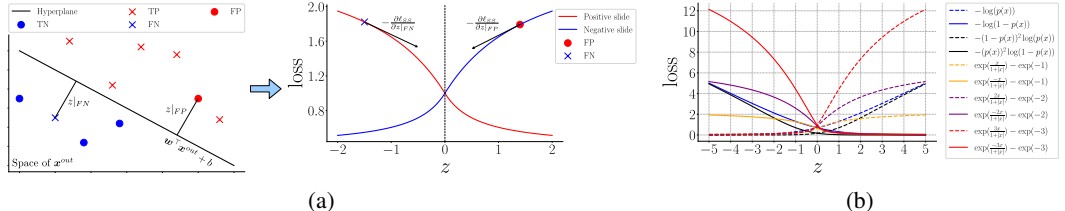

(a)                                                                 (b)

Figure 2: Conceptual workflow of the proposed steep slope loss (a) and graph comparison between the proposed loss and the conventional losses (b). In (b), the cross entropy loss and focal loss are plotted in blue and black, respectively. The TCP confidence loss is a square error and varies with the classification confidence. Therefore, it is not plotted here.

## 3  Methodology

In this section, we first introduce the overall learning framework for predicting trustworthiness. Then, we narrow down to the proposed steep slope loss function. At last, we provide the generalization bound that is related to the proposed steep slope loss function.

### 3.1  Overall Design

Corbière et al. [9] provide a good learning scheme for predicting trustworthiness. Briefly, it first trains a classifier with the training samples. Then, the classifier is frozen and the confidence network (i.e., trustworthiness predictor) is trained (or fine-tuned) with the training samples. In general, we follows this learning scheme.

This work focuses on the trustworthiness on the predictions yielded by the publicly available pre-trained classifiers, i.e., ViT [7] and ResNet [5]. We use the pre-trained backbones as the backbones of the oracles for general purposes. In this sense, the set of the oracle's parameters can be split into two parts, one is related to the backbone and the other one is related to the head, i.e., $\theta = \{\theta_{backbone}, \theta_{head}\}$. $\theta_{backbone}$ is used to generated the intermediate feature $\boldsymbol{x}^{out}$ and $\theta_{head} = \{\boldsymbol{w}, b\}$ are usually the weights of a linear function to generate the discriminative feature $z = \boldsymbol{w}^{\top}\boldsymbol{x}^{out} + b$. With the classifier, the oracle, a pre-defined loss, and the training samples, we can optimize problem (3) to find the optimal parameters for the oracle.

### 3.2  Steep Slope Loss

The conceptual workflow of the proposed steep slope loss is shown in Fig. 2a. The core idea is that we exploit the graph characteristics of the exponential function and the softsign function to establish two slides such that the features $z$ w.r.t. the positive class ride down the positive slide to the right bottom and the features $z$ w.r.t. the negative class ride down the negative slide to the left bottom. $z$ is defined as the signed distance to the hyperplane (i.e., oracle head). Given an image $\boldsymbol{x}$, $z = f_\theta(\boldsymbol{x})$ can be broken down into

$$z = \frac{\boldsymbol{w}^{\top}\boldsymbol{x}^{out} + b}{\|\boldsymbol{w}\|}, \quad \boldsymbol{x}^{out} = f_{\theta_{backbone}}(\boldsymbol{x}). \tag{7}$$

The signed distance to the hyperplane has a geometric interpretation: its sign indicates in which half-space $\boldsymbol{x}^{out}$ is and its absolute value indicates how far $\boldsymbol{x}^{out}$ is away from the hyperplane.

It is desired that the signed distance of $\boldsymbol{x}^{out}$ with ground-truth label $o = 1$ ($o = 0$) tends towards $+\infty$ ($-\infty$) as much as possible. To this end, we define the steep slope (SS) loss function as follows

$$\ell_{SS}(f_\theta(\boldsymbol{x}), o) = o \cdot \underbrace{\left(\exp\left(\frac{\alpha^+ z}{1 + |z|}\right) - \exp(-\alpha^+)\right)}_{\text{Positive slide}} + (1 - o) \cdot \underbrace{\left(\exp\left(\frac{-\alpha^- z}{1 + |z|}\right) - \exp(-\alpha^-)\right)}_{\text{Negative slide}} \tag{8}$$

where $\alpha^+, \alpha^- \in \mathbb{R}^+$ control the slope of the positive slide and the negative slide, respectively. If $z$ w.r.t. the positive class is on the left-hand side of $z = 0$, minimizing the loss would push the point on the hill down to the bottom, i.e., the long tail region indicating the well-classified region. Similarly, $z$

w.r.t. the negative class would undergo a similar process. $\exp(-\alpha^+)$ and $\exp(-\alpha^-)$ are vertical shifts for the positive and negative slides such that $\ell_{ss}$ has a minimum value 0. Note that the proposed steep slope loss is in the range $[0, \text{maximum}\{\exp(\alpha^+) - \exp(-\alpha^+), \exp(\alpha^-) - \exp(-\alpha^-)\}]$, whereas the cross entropy loss and the focal loss are in the range $[0, +\infty)$. The proposed steep slope loss can work with the output of the linear function as well. This is because the signed distance and the output of the linear function have a proportional relationship with each other, i.e., $\frac{\boldsymbol{w}^\top \boldsymbol{x}^{out} + b}{\|\boldsymbol{w}\|} \propto \boldsymbol{w}^\top \boldsymbol{x}^{out} + b$.

Essentially, as shown in Fig. 2b, the cross entropy loss, focal loss, and steep slope loss work in a similar manner to encourage $z$ w.r.t. the positive class to move the right-hand side of the positive threshold 0 and encourage $z$ w.r.t. the negative class to move the left-hand side of the negative threshold 0, which is analogous to the sliding motion. The proposed steep slope loss is more tractable to control the steepness of slopes than the cross entropy loss and focal loss. This leads to an effective learning to yield discriminative feature for predicting trustworthiness.

### 3.3 Generalization Bound

With the proposed steep slope loss, we are interested in the generalization bound of trustworthiness predictors. For simplicity, we simplify a trustworthiness predictor as $f \in \mathcal{F}$, where $\mathcal{F}$ is a finite hypothesis set. The risk of predicting trustworthiness is defined as $\mathcal{R}(f) = \mathbb{E}_{(\boldsymbol{x},y) \sim P}[\ell_{SS}(f(\boldsymbol{x}), o)]$, where $P$ is the underlying joint distribution of $(\boldsymbol{x}, o)$. As $P$ is inaccessible, a common practice is to use empirical risk minimization (ERM) to approximate the risk [19], i.e., $\hat{\mathcal{R}}_D(f) = \frac{1}{|D|} \sum_{i=1}^{|D|} \ell_{SS}(f(\boldsymbol{x}_i), o_i)$.

The following theorem provides an insight into the correlation between the generalization bound and the loss function in the learning task for predicting trustworthiness.

**Theorem 3.1.** *Denote maximum*$\{\exp(\alpha^+) - \exp(-\alpha^+), \exp(\alpha^-) - \exp(-\alpha^-)\}$ *as* $\ell_{SS}^{max}$. $\ell_{SS} \in [0, \ell_{SS}^{max}]$. *Assume* $\mathcal{F}$ *is a finite hypothesis set, for any* $\delta > 0$, *with probability at least* $1 - \delta$, *the following inequality holds for all* $f \in \mathcal{F}$:

$$|\mathcal{R}(f) - \hat{\mathcal{R}}_D(f)| \leq \ell_{SS}^{max} \sqrt{\frac{\log |\mathcal{F}| + \log \frac{2}{\delta}}{2|D|}}$$

The proof sketch is similar to the generalization bound provided in [20] and the detailed proof is provided in the appendix A.

A desired characteristic of the proposed steep slope loss is that it is in a certain range determined by $\alpha^+$ and $\alpha^-$, as discussed in Section 3.2. This leads to the generalization bound shown in Theorem 3.1. The theorem implies that given a generic classifier, as the number of training samples increases, the empirical risk would be close to the true risk with a certain probability. On the other hand, the cross entropy loss, focal loss, and TCP loss are not capped in a certain range. They do not fit under Hoeffding's inequality to derive the generalization bounds.

## 4 Related Work

**Loss Function**. Loss function is the key to search for optimal parameters and has been extensively studied in a range of learning tasks [9, 18, 21, 22]. Specifically, the cross entropy loss may be the most widely-used loss for classification [21], however, it is not optimal for all cases. Lin et al. [18] propose the focal loss to down-weight the loss w.r.t. well-classified examples and focus on minimizing the loss w.r.t. misclassified examples by reshaping the cross entropy loss function. Liu et al. [22] introduce the large margin softmax loss that is based on cosine similarity between the weight vectors and the feature vectors. Nevertheless, it requires more hyperparameters and entangles with the linear function in the last layer of the network. Similar to the idea of focal loss, the proposed steep slope loss is flexible to reshape the function graphs to improve class imbalance problem. TCP's confidence loss is used to train the confidence network for predicting trustworthiness [9]. We adopt TCP as a baseline.

**Trustworthiness Prediction**. Understanding trustworthiness of a classifier has been studied in the past decade [8, 9, 13, 23, 24]. Hendrycks and Gimpel [13] aim to detect the misclassified examples according to maximum class probability. This method depends on the ranking of confidence scores and is proved to be unreliable [8]. Monte Carlo dropout (MCDropout) intends to understand trustworthiness through the lens of uncertainty estimation [23]. However, it is difficult to distinguish

the trustworthy prediction from the incorrect but overconfident predictions [9]. Jiang et al. [8] propose a confidence measure named trust score, which is based on the distance between the classifier and a modified nearest-neighbor classifier on the test examples. The shortcomings of the method are the poor scalability and computationally expensive. Recently, instead of implementing a standalone trustworthiness predictor, Moon et al. [24] use the classifier's weights to compute the correctness ranking. The correctness ranking is taken into account in the loss function such that the classification accuracy is improved. This method relies on pairs of training samples for computing the ranking. Moreover, it is unclear if the classifier improved by the correctness ranking outperforms the pre-trained classifier when applying the method on large-scale datasets, e.g., ImageNet. In contrast, the learning scheme to train the trustworthiness predictor (i.e., confidence network) in [9] is standard and does not affect the classifiers. Its efficacy has been verified on small-scale datasets. In this work, we follows the learning scheme used in [9] and focus on predicting trustworthiness on complex dataset.

**Imbalanced Classification**. In classification task, the ratio of correct predictions to incorrect predictions is expected to be large due to the advance in deep learning methods. This is aligned with the nature of imbalanced classification [25, 26, 27, 28, 29], which generally employ re-sampling and re-weighting strategies to solve the problem. However, predicting trustworthiness is different from imbalanced classification as it is not aware of the visual concepts but the correctness of the predictions, whereas the classification task relies on the visual concepts. Specifically, the re-sampling based methods [25, 26, 29] may not be suitable for the problem of predicting trustworthiness. It is difficult to determine if a sample is under-represented (over-represented) and should be over-sampled (under-sampled). The re-weighting based methods [26, 28] can be applied to any generic loss functions, but they hinge on some hypothesis related to imbalanced data characteristics. For instance, the class-balanced loss [28] is based on the effective number w.r.t. a class, which presumes number of samples is known. However, the number of samples w.r.t. a class is not always assessible, e.g., in the online learning scheme [30, 31]. Instead of assuming some hypothesis, we follow the standard learning scheme for predicting trustworthiness [8, 9].

# 5    Experiment & Analysis

In this section, we first introduce the experimental set-up. Then, we report the performances of baselines and the proposed steep slope loss on ImageNet, followed by comprehensive analyses.

**Experimental Set-Up**. We use ViT B/16 [7] and ResNet-50 [5] as the classifiers, and the respective backbones are used as the oracles' backbones. We denote the combination of oracles and classifiers as $\langle O, C \rangle$. There are four combinations in total, i.e., $\langle ViT, ViT \rangle$, $\langle ViT, RSN \rangle$, $\langle RSN, ViT \rangle$, and $\langle RSN, RSN \rangle$, where RSN stands for ResNet. In this work, we adopt three baselines, i.e., the cross entropy loss [21], focal loss [18], and TCP confidence loss [9], for comparison purposes.

The experiment is conducted on ImageNet [11], which consists of 1.2 million labeled training images and 50000 labeled validation images. It has 1000 visual concepts. Similar to the learning scheme in [9], the oracle is trained with training samples and evaluated on the validation set. During the training process of the oracle, the classifier works in the evaluation mode so training the oracle would not affect the parameters of the classifier. Moreover, we conduct the analyses about how well the learned oracle generalizes to the images in the unseen domains. To this end, we apply the widely-used style transfer method [32] and the functional adversarial attack method [33] to generate two variants of the validation set, i.e., stylized validation set and adversarial validation set. Also, we adopt ImageNet-C [34] for evaluation, which is used for evaluating robustness to common corruptions.

The oracle's backbone is initialized by the pre-trained classifier's backbone and trained by fine-tuning using training samples and the trained classifier. Training the oracles with all the loss functions uses the same hyperparameters, such as learning rate, weight decay, momentum, batch size, etc. The details for the training process and the implementation are provided in Appendix B.

For the focal loss, we follow [18] to use $\gamma = 2$, which leads to the best performance for object detection. For the proposed loss, we use $\alpha^+ = 1$ and $\alpha^- = 3$ for the oracle that is based on ViT's backbone, while we use $\alpha^+ = 2$ and $\alpha^- = 5$ for the oracle that is based on ResNet's backbone.

Following [9], we use FPR-95%-TPR, AUPR-Error, AUPR-Success, and AUC as the metrics. FPR-95%-TPR is the false positive rate (FPR) when true positive rate (TPR) is equal to 95%. AUPR is the area under the precision-recall curve. Specifically, AUPR-Success considers the correct prediction

Table 1: Performance on the ImageNet validation set. The mean and the standard deviation of scores are computed over three runs. The oracles are trained with the ImageNet training samples. The classifier is used in the evaluation mode. Acc is the classification accuracy and is helpful to understand the proportion of correct predictions. *SS* stands for the proposed steep slope loss.

| $\langle O, C \rangle$ | Loss | Acc↑ | FPR-95%-TPR↓ | AUPR-Error↑ | AUPR-Success↑ | AUC↑ | TPR↑ | TNR↑ |
|---|---|---|---|---|---|---|---|---|
| $\langle$ViT, ViT$\rangle$ | CE | 83.90 | 93.01±0.17 | **15.80**±0.56 | 84.25±0.57 | 51.62±0.86 | **99.99**±0.01 | 0.02±0.02 |
| | Focal [18] | 83.90 | 93.37±0.52 | 15.31±0.44 | 84.76±0.50 | 52.38±0.77 | 99.15±0.14 | 1.35±0.22 |
| | TCP [9] | 83.90 | 88.38±0.23 | 12.96±0.10 | 87.63±0.15 | 60.14±0.47 | 99.73±0.02 | 0.00±0.00 |
| | SS | 83.90 | **80.48**±0.66 | 10.26±0.03 | **93.01**±0.10 | **73.68**±0.27 | 87.52±0.95 | **38.27**±2.48 |
| $\langle$ViT, RSN$\rangle$ | CE | 68.72 | 93.43±0.28 | 30.90±0.35 | 69.13±0.36 | 51.24±0.63 | **99.90**±0.04 | 0.20±0.00 |
| | Focal [18] | 68.72 | 93.94±0.51 | **30.97**±0.36 | 69.07±0.35 | 51.26±0.62 | 93.66±0.29 | 7.71±0.53 |
| | TCP [9] | 68.72 | 83.55±0.70 | 23.56±0.47 | 79.04±0.91 | 66.23±1.02 | 94.25±0.96 | 0.00±0.00 |
| | SS | 68.72 | **77.89**±0.39 | 20.91±0.05 | **85.39**±0.16 | **74.31**±0.21 | 68.32±0.41 | **67.53**±0.62 |
| $\langle$RSN, ViT$\rangle$ | CE | 83.90 | 93.29±0.53 | 14.74±0.17 | 85.40±0.20 | 53.43±0.28 | **100.00**±0.00 | 0.00±0.00 |
| | Focal [18] | 83.90 | 94.60±0.53 | **14.98**±0.21 | 85.13±0.24 | 52.37±0.51 | **100.00**±0.00 | 0.00±0.00 |
| | TCP [9] | 83.90 | 91.93±0.49 | 14.12±0.12 | 86.12±0.15 | 55.55±0.46 | **100.00**±0.00 | 0.00±0.00 |
| | SS | 83.90 | **88.70**±0.08 | 11.69±0.04 | **90.01**±0.10 | **64.34**±0.16 | 96.20±0.73 | **9.00**±1.32 |
| $\langle$RSN, RSN$\rangle$ | CE | 68.72 | 94.84±0.27 | 29.41±0.18 | 70.79±0.19 | 52.36±0.41 | **100.00**±0.00 | 0.00±0.00 |
| | Focal [18] | 68.72 | 95.16±0.19 | **29.92**±0.38 | 70.23±0.44 | 51.43±0.50 | 99.86±0.05 | 0.08±0.03 |
| | TCP [9] | 68.72 | 88.81±0.24 | 24.46±0.12 | 77.79±0.29 | 62.73±0.14 | 99.23±0.14 | 0.00±0.00 |
| | SS | 68.72 | **86.21**±0.44 | 22.53±0.03 | **81.88**±0.10 | **67.92**±0.11 | 79.20±2.50 | **42.09**±3.77 |

as the positive class, whereas AUPR-Error considers the incorrect prediction as the positive class. AUC is the area under the receiver operating characteristic curve, which is the plot of TPR versus FPR. Moreover, we use TPR and true negative rate (TNR) as additional metrics because they assess overfitting issue, e.g., TPR=100% and TNR=0% imply that the trustworthiness predictor is prone to view all the incorrect predictions to be trustworthy.

**Performance on Large-Scale Dataset**. The result on ImageNet are reported in Table 1. We have two key observations. Firstly, training with the cross entropy loss, focal loss, and TCP confidence loss lead to overfitting the imbalanced training samples, i.e., the dominance of trustworthy predictions. Specifically, TPR is higher than 99% while TNR is less than 1% in all cases. Secondly, the performance of predicting trustworthiness is contingent on both the oracle and the classifier. When a high-performance model (i.e., ViT) is used as the oracle and a relatively low-performance model (i.e., ResNet) is used as the classifier, cross entropy loss and focal loss achieve higher TNRs than the loss functions with the other combinations. In contrast, the two losses with $\langle$ResNet, ViT$\rangle$ lead to the lowest TNRs (i.e., 0%).

Despite the combinations of oracles and classifiers, the proposed steep slope loss can achieve significantly higher TNRs than using the other loss functions, while it achieves desirable performance on FPR-95%-TPR, AUPR-Success, and AUC. This verifies that the proposed loss is effective to improve the generalizability for predicting trustworthiness. Note that the scores of AUPR-Error and TPR yielded by the proposed loss are lower than that of the other loss functions. Recall that AUPR-Error aims to inspect how easy to detect failures and depends on the negated trustworthiness confidences w.r.t. incorrect predictions [9]. The AUPR-Error correlates to TPR and TNR. When TPR is close to 100% and TNR is close to 0%, it indicates the oracle is prone to view all the predictions to be trustworthy. In other words, almost all the trustworthiness confidences are on the right-hand side of $p(o = 1|\theta, \boldsymbol{x}) = 0.5$. Consequently, when taking the incorrect prediction as the positive class, the negated confidences are smaller than -0.5. On the other hand, the oracle trained with the proposed loss intends to yield the ones w.r.t. incorrect predictions that are smaller than 0.5. In general, the negated confidences w.r.t. incorrect predictions are greater than the ones yielded by the other loss functions. In summary, a high TPR score and a low TNR score leads to a high AUPR-Error.

To intuitively understand the effects of all the loss functions, we plot the histograms of trustworthiness confidences w.r.t. true positive (TP), false positive (FP), true negative (TN), and false negative (FN) in Fig. 3. The result confirms that the oracles trained with the baseline loss functions are prone to predict overconfident trustworthiness for incorrect predictions, while the oracles trained with the proposed loss can properly predict trustworthiness for incorrect predictions.

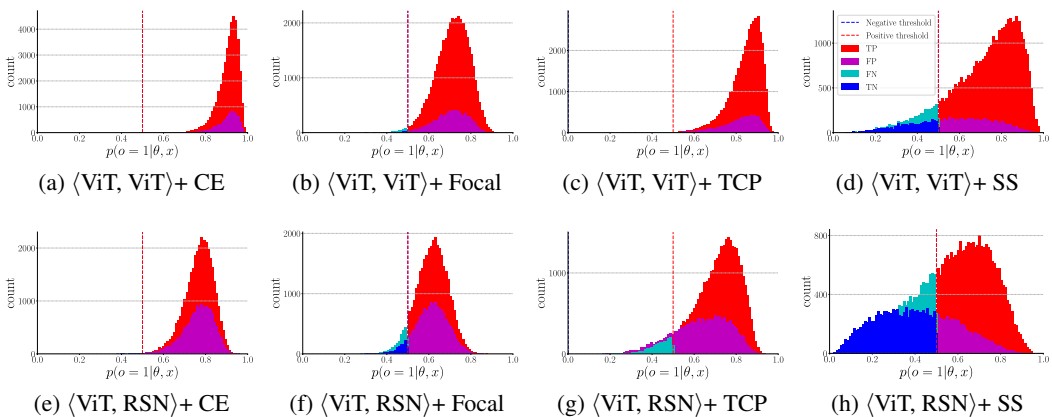

| (a) $\langle$ViT, ViT$\rangle$+ CE | (b) $\langle$ViT, ViT$\rangle$+ Focal | (c) $\langle$ViT, ViT$\rangle$+ TCP | (d) $\langle$ViT, ViT$\rangle$+ SS |
|---|---|---|---|
| (e) $\langle$ViT, RSN$\rangle$+ CE | (f) $\langle$ViT, RSN$\rangle$+ Focal | (g) $\langle$ViT, RSN$\rangle$+ TCP | (h) $\langle$ViT, RSN$\rangle$+ SS |

Figure 3: Histograms of trustworthiness confidences w.r.t. all the loss functions on the ImageNet validation set. The oracles that are used to generate the confidences are the ones used in Table 1. The histograms generated with $\langle$RSN, ViT$\rangle$ and $\langle$RSN, RSN$\rangle$ are provided in Appendix D.

Table 2: Performance on MNIST and CIFAR-10.

| Dataset | Loss | Acc↑ | FPR-95%-TPR↓ | AUPR-Error↑ | AUPR-Success↑ | AUC↑ | TPR↑ | TNR↑ |
|---|---|---|---|---|---|---|---|---|
| MNIST | MCP [13] | 99.10 | 5.56 | 35.05 | **99.99** | 98.63 | 99.89 | **8.89** |
| | MCDropout [23] | 99.10 | 5.26 | 38.50 | **99.99** | 98.65 | - | - |
| | TrustScore [8] | 99.10 | 10.00 | 35.88 | 99.98 | 98.20 | - | - |
| | TCP [9] | 99.10 | 3.33 | **45.89** | **99.99** | 98.82 | 99.71 | 0.00 |
| | TCP† | 99.10 | 3.33 | 45.88 | **99.99** | 98.82 | 99.72 | 0.00 |
| | SS | 99.10 | **2.22** | 40.86 | **99.99** | **98.83** | **100.00** | 0.00 |
| CIFAR-10 | MCP [13] | 92.19 | 47.50 | 45.36 | 99.19 | 91.53 | 99.64 | 6.66 |
| | MCDropout [23] | 92.19 | 49.02 | 46.40 | **99.27** | 92.08 | - | - |
| | TrustScore [8] | 92.19 | 55.70 | 38.10 | 98.76 | 88.47 | - | - |
| | TCP [9] | 92.19 | 44.94 | 49.94 | 99.24 | 92.12 | **99.77** | 0.00 |
| | TCP† | 92.19 | 45.07 | 49.89 | 99.24 | 92.12 | 97.88 | 0.00 |
| | SS | 92.19 | **44.69** | **50.28** | 99.26 | **92.22** | 98.46 | **28.04** |

**Separability between Distributions of Correct Predictions and Incorrect Predictions**. As observed in Fig. 3, the confidences w.r.t. correct and incorrect predictions follow Gaussian-like distributions. Hence, we can compute the separability between the distributions of correct and incorrect predictions from a probabilistic perspective. Given the distribution of correct predictions $\mathcal{N}_1(\mu_1, \sigma_1^2)$ and the distribution of correct predictions $\mathcal{N}_2(\mu_2, \sigma_2^2)$, we use the average Kullback–Leibler (KL) divergence $\bar{d}_{KL}(\mathcal{N}_1, \mathcal{N}_2)$ [35] and Bhattacharyya distance $d_B(\mathcal{N}_1, \mathcal{N}_2)$ [36] to measure the separability. More details and the quantitative results are reported in Appendix G. In short, the proposed loss leads to larger separability than the baseline loss functions. This implies that the proposed loss is more effective to differentiate incorrect predictions from correct predictions.

**Performance on Small-Scale Datasets**. We also provide comparative experimental results on small-scale datasets, i.e., MNIST [10] and CIFAR-10 [37]. The results are reported in Table 2. The proposed loss outperforms TCP† on metric FPR-95%-TPR on both MNIST and CIFAR-10, and additionally achieved good performance on metrics AUPR-Error and TNR on CIFAR-10. This shows the proposed loss is able to adapt to relatively simple data. More details can be found in Appendix B.1.

**Generalization to Unseen Domains**. In practice, the oracle may run into the data in the domains that are different from the ones of training samples. Thus, it is interesting to find out how well the learned oracles generalize to the unseen domain data. Using the oracles trained with the ImageNet training set (i.e., the ones used in Table 1), we evaluate it on the stylized ImageNet validation set [32], adversarial ImageNet validation set [33], and corrupted ImageNet validation set [34]. $\langle$ViT, ViT$\rangle$ is used in the experiment.

The results on the stylized ImageNet, adversarial ImageNet, and ImageNet-C are reported in Table 3, More results on ImageNet-C are reported in Table A1. As all unseen domains are different from the

Table 3: Performance on the stylized ImageNet validation set, the adversarial ImageNet validation set, and one (Defocus blur) of validation sets in ImageNet-C. Defocus blus is at at the highest level of severity. ⟨ViT, ViT⟩ is used in the experiment and the domains of the two validation sets are different from the one of the training set that is used for training the oracle. The corresponding histograms are available in Appendix D. More results on ImageNet-C can be found in Table A1.

| Dataset | Loss | Acc↑ | FPR-95%-TPR↓ | AUPR-Error↑ | AUPR-Success↑ | AUC↑ | TPR↑ | TNR↑ |
|---|---|---|---|---|---|---|---|---|
| Stylized [32] | CE | 15.94 | 95.52 | 84.18 | 15.86 | 49.07 | **99.99** | 0.02 |
| | Focal [18] | 15.94 | 95.96 | **85.90** | 14.30 | 46.01 | 99.71 | 0.25 |
| | TCP [9] | 15.94 | 93.42 | 80.17 | 21.25 | 57.29 | 99.27 | 0.00 |
| | SS | 15.94 | **89.38** | 75.08 | **34.39** | **67.68** | 44.42 | **81.22** |
| Adversarial [33] | CE | 6.14 | 94.35 | **93.70** | 6.32 | 51.28 | **99.97** | 0.06 |
| | Focal [18] | 6.15 | 93.67 | 93.48 | 6.56 | 52.39 | 99.06 | 1.43 |
| | TCP [9] | 6.11 | 93.94 | 92.77 | 7.55 | 55.81 | 99.71 | 0.00 |
| | SS | 6.16 | **90.07** | 90.09 | **13.07** | **65.36** | 87.10 | **24.33** |
| Defocus blur [34] | CE | 31.83 | 94.46 | **68.56** | 31.47 | 50.13 | **99.15** | 1.07 |
| | Focal [18] | 31.83 | 94.98 | 66.87 | 33.24 | 51.28 | 96.70 | 3.26 |
| | TCP [9] | 31.83 | 93.50 | 64.67 | 36.05 | 54.27 | 96.71 | 4.35 |
| | SS | 31.83 | **90.18** | 57.95 | **48.80** | **64.34** | 77.79 | **37.29** |

one of the training set, the classification accuracies are much lower than the ones in Table 1. The adversarial validation set is also more challenging than the stylized validation set and the corrupted validation set. As a result, the difficulty affects the scores across all metrics. The oracles trained with the baseline loss functions are still prone to recognize the incorrect prediction to be trustworthy. The proposed loss consistently improves the performance on FPR-95%-TPR, AUPR-Sucess, AUC, and TNR. Note that the adversarial perturbations are computed on the fly [33]. Instead of truncating the sensitive pixel values and saving into the images files, we follow the experimental settings in [33] to evaluate the oracles on the fly. Hence, the classification accuracies w.r.t. various loss function are slightly different but are stably around 6.14%.

**Selective Risk Analysis**. Risk-coverage curve is an important technique for analyzing trustworthiness through the lens of the rejection mechanism in the classification task [9, 38]. In the context of predicting trustworthiness, selective risk is the empirical loss that takes into account the decisions, i.e., to trust or not to trust the prediction. Correspondingly, coverage is the probability mass of the non-rejected region. As can see in Fig. 4a, the proposed loss leads to significantly lower risks, compared to the other loss functions. We present the risk-coverage curves w.r.t. all the combinations of oracles and classifiers in Appendix E. They consistently exhibit similar pattern.

**Ablation Study**. In contrast to the compared loss functions, the proposed loss has more hyperparameters to be determined, i.e., $\alpha^+$ and $\alpha^-$. As the proportion of correct predictions is usually larger than that of incorrect predictions, we would prioritize $\alpha^-$ over $\alpha^+$ such that the oracle is able to recognize a certain amount of incorrect predictions. In other words, we first search for $\alpha^-$ by freezing $\alpha^+$, and then freeze $\alpha^-$ and search for $\alpha^+$. Fig. 4b and 4c show how the loss, TPR, and TNR vary with various $\alpha^-$. In this analysis, the combination ⟨ViT, ViT⟩ is used and $\alpha^+ = 1$. We can see that

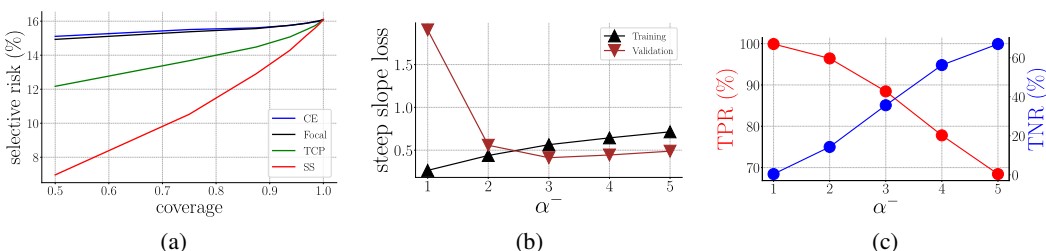

(a)  (b)  (c)

Figure 4: Analyses based on ⟨ViT, ViT⟩. (a) are the curves of risk vs. coverage. Selective risk represents the percentage of errors in the remaining validation set for a given coverage. (b) are the curves of loss vs. $\alpha^-$. (c) are TPR and TNR against various $\alpha^-$.

$\alpha^- = 3$ achieves the optimal trade-off between TPR and TNR. We follow a similar search strategy to determine $\alpha^+ = 2$ and $\alpha^- = 5$ for training the oracle with ResNet backbone.

**Effects of Using** $z = \boldsymbol{w}^\top \boldsymbol{x}^{out} + b$. Using the signed distance as $z$, i.e., $z = \frac{\boldsymbol{w}^\top \boldsymbol{x}^{out} + b}{\|\boldsymbol{w}\|}$, has a geometric interpretation as shown in Fig. 2a. However, the main-stream models [5, 6, 7] use $z = \boldsymbol{w}^\top \boldsymbol{x}^{out} + b$. Therefore, we provide the corresponding results in appendix F, which are generated by the proposed loss taking the output of the linear function as input. In comparison with the results of using $z = \frac{\boldsymbol{w}^\top \boldsymbol{x}^{out} + b}{\|\boldsymbol{w}\|}$, using $z = \boldsymbol{w}^\top \boldsymbol{x}^{out} + b$ yields comparable scores of FPR-95%-TPR, AUPR-Error, AUPR-Success, and AUC. Also, TPR and TNR are moderately different between $z = \frac{\boldsymbol{w}^\top \boldsymbol{x}^{out} + b}{\|\boldsymbol{w}\|}$ and $z = \boldsymbol{w}^\top \boldsymbol{x}^{out} + b$, when $\alpha^+$ and $\alpha^-$ are fixed. This implies that TPR and TNR are sensitive to $\|\boldsymbol{w}\|$. This is because the normalization by $\|w\|$ would make $z$ more dispersed in value than the variant without normalization. In other words, the normalization leads to long-tailed distributions while no normalization leads to short-tailed distributions. Given the same threshold, TNR (TPR) is determined by the location of the distribution of negative (positive) examples and the extent of short/long tails. Our analysis on the histograms generated with and without $\|w\|$ normalization verifies this point.

**Steep Slope Loss vs. Class-Balanced Loss**. We compare the proposed loss to the class-balanced loss [28], which is based on a re-weighting strategy. The results are reported in Appendix H. Overall, the proposed loss outperforms the class-balanced loss, which implies that the imbalance characteristics of predicting trustworthiness is different from that of imbalanced data classification.

# 6  Conclusion

In this work, we study the problem of predicting trustworthiness on a large-scale dataset. We observe that the oracle, i.e., trustworthiness predictor, trained with the cross entropy loss, focal loss, and TCP confidence loss lean towards viewing incorrect predictions to be trustworthy due to overfitting. To improve the generalizability of the oracles, we propose the steep slope loss that encourages the features w.r.t. correct predictions and incorrect predictions to be separated from each other. We evaluate the proposed loss on ImageNet through the lens of the trustworthiness metrics, selective classification metric, and separability of distributions, respectively. Experimental results show that the proposed loss is effective in improving the generalizability of trustworthiness predictors.

# 7  Societal Impact

Learning high-accuracy models is a long-standing goal in machine learning. Nevertheless, due to the complexity of real-world data, there is still a gap between state-of-the-art classification models and a perfect one. Hence, there is a critical need to understand the trustworthiness of classification models, i.e., differentiate correct predictions and incorrect predictions, in order to safely and effectively apply the models in real-world tasks. Models with the proposed loss achieve considerably better performance with various trustworthiness metrics. They also show generalizability with both in-distribution and out-of-distribution images at a large scale. In addition, while existing trustworthiness models focus on a high TPR and tend to view all incorrect predictions to be trustworthy (i.e., TNR close to 0), false positives may lead to high consequent cost in some real-world scenarios such as critical applications in medicine (e.g., a false positive leading to unnecessary and invasive tests or treatment such as biopsies or surgery, or harmful side effects in medicine) and security (e.g., counter terrorism). It is thus of great importance for a trustworthiness model to flexibly trade off between TPR and TNR. To this end, the proposed loss allows the underlying distributions of positive and negative examples to be more separable, enabling a more effective trade-off between them.

**Acknowledgments and Disclosure of Funding**

This research was funded in part by the NSF under Grants 1908711, 1849107, and in part supported by the National Research Foundation, Singapore under its Strategic Capability Research Centres Funding Initiative. Any opinions, findings and conclusions or recommendations expressed in this material are those of the author(s) and do not reflect the views of National Research Foundation, Singapore.

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
