# Appendix for Learning to Predict Trustworthiness with Steep Slope Loss

**Yan Luo**[†], **Yongkang Wong**[‡], **Mohan S Kankanhalli**[‡], **Qi Zhao**[†]

[†] Department of Computer Science & Engineering, University of Minnesota
[‡] School of Computing, National University of Singapore

luoxx648@umn.edu, yongkang.wong@nus.edu.sg, mohan@comp.nus.edu.sg, qzhao@cs.umn.edu

## A   Generalization Bound

**Theorem.** *Denote maximum$\{\exp(\alpha^+) - \exp(-\alpha^+), \exp(\alpha^-) - \exp(-\alpha^-)\}$ as $\ell_{SS}^{max}$. $\ell_{SS} \in [0, \ell_{SS}^{max}]$. Assume $\mathcal{F}$ is a finite hypothesis set, for any $\delta > 0$, with probability at least $1 - \delta$, the following inequality holds for all $f \in \mathcal{F}$:*

$$|\mathcal{R}(f) - \hat{\mathcal{R}}_D(f)| \leq \ell_{SS}^{max}\sqrt{\frac{\log|\mathcal{F}| + \log\frac{2}{\delta}}{2|D|}}$$

*Proof.* The proof sketch is similar to the generalization bound provided in [1]. By the union bound, given an error $\xi$, we have

$$p[\sup_{f \in \mathcal{F}}|R(f) - \hat{R}(f)| > \xi] \leq \sum_{f \in \mathcal{F}} p[|R(f) - \hat{R}(f)| > \xi].$$

By Hoeffding's bound, we have

$$\sum_{f \in \mathcal{F}} p[|\mathcal{R}(f) - \hat{\mathcal{R}}(f)| > \xi] \leq 2|\mathcal{F}|\exp\left(-\frac{2|D|\xi^2}{(\ell_{SS}^{max})^2}\right).$$

Due to the probability definition, $2|\mathcal{F}|\exp(-\frac{2|D|\xi^2}{(\ell_{SS}^{max})^2}) = \delta$. Considering $\xi$ is a function of other variables, we can rearrange it as $\xi = \ell_{SS}^{max}\sqrt{\frac{\log|\mathcal{F}| + \log\frac{2}{\delta}}{2|D|}}$. Since we know $p[|\mathcal{R}(f) - \hat{\mathcal{R}}(f)| > \xi]$ is with probability at most $\delta$, it can be inferred that $p[|\mathcal{R}(f) - \hat{\mathcal{R}}(f)| <= \xi]$ is at least $1 - \delta$.  ☐

## B   Experimental Set-Up

The ViT (i.e., ViT Base/16) used in this work is implemented in the ASYML project[1], which is based on PyTorch. The pre-trained weights are the same as the original pre-trained weights[2]. On the other hand, the pre-trained ResNet (i.e., ResNet-50) is provided in PyTorch[3]. For the analyses, we use the official implementation[4] of the TCP confidence loss [2] and the PyTorch implementation[5] of the class-balanced loss [3].

---

[1] https://github.com/asyml/vision-transformer-pytorch
[2] https://github.com/google-research/vision_transformer
[3] https://pytorch.org/vision/stable/models.html
[4] https://github.com/valeoai/ConfidNet
[5] https://github.com/vandit15/Class-balanced-loss-pytorch

35th Conference on Neural Information Processing Systems (NeurIPS 2021).

We use the training scheme implemented by ASYML and tune the hyperparameters such that the oracles are trained with the cross entropy loss and focal loss to produce the best performance among multiple trials. Then, we fix the set of hyperparameters for the TCP confidence loss and the proposed loss. Each combination of classifiers and oracles undergoes the same training scheme. Specifically, the stochastic gradient descent (SGD) optimization method is used with initial learning rate 1e-5, weight decay 0, and momentum 0.05 for optimizing the learning problem. The 1-cycle learning rate policy applies at each learning step. The batch size is fixed to 40 as ViT would make the full use of four 12 GB GPUs with 40 images. To stimulate a challenging and practically useful environment, we train the oracle in only one epoch, rather than multiple epochs. All the three baseline loss functions and the proposed loss use the same hyperparameters and undergo the same experimental protocol for training the oracle. The code is implemented in Python 3.8.5 with PyTorch 1.7.1 [4] and is tested under Ubuntu 18.04 with four NVIDIA GTX 1080 Ti graphics cards in a standalone machine.

We run the experiments three times with random seeds and report the means and the standard deviations of scores in Table 1. For the other experiments or analyses, we run one time.

### B.1  Implementation Details for Small-Scale Datasets

The resulting results are reported in Table 2. The experiment is based on the official implementation[6] of [2]. The implementation provides the pre-trained models on MNIST and CIFAR-10. We fine-tune the pre-trained models with the proposed steep slope loss. For comparison purposes, we also fine-tune the pre-trained with the TCP confidence loss (i.e., $TCP\dagger$), where the experimental settings of the fine-tuning process are the same as the ones of the fine-tuning process with the proposed steep slope loss. The proposed loss use $\alpha^+ = 10$ and $\alpha^- = 6$ on MNIST, and $\alpha^+ = 1$ and $\alpha^- = 1$ on CIFAR-10.

## C  License of Assets

MNIST [5] is made available under the terms of the Creative Commons Attribution-Share Alike 3.0 license, while ImageNet [6] is licensed under the BSD 3-Clause "New" or "Revised" License.

PyTorch [4] is available under a BSD-style license. The official ViT [7] implementation is licensed under the Apache-2.0 License, while the implementation of ViT is licensed under the Apache-2.0 License. The code of TCP [2] is licensed under the Apache License. The PyTorch version of class balanced loss [3] is licensed under the MIT License.

We make our code and pre-trained oracles publicly available via `https://github.com/luoyan407/predict_trustworthiness` with the MIT License.

## D  Experimental Result

As shown in Table 1 and discussed in the experiment section, the proposed loss consistently improves the performance on metrics FPR-95%-TPR, AUPR-Success, AUC, and TNR while the corresponding variances are comparable to the other loss functions.

We plot all the histograms in Fig. A1 and Fig. A2 that correspond to Table 1 and Table 3, respectively. Ideally, we hope that all the confidences w.r.t. the positive class are on the right-hand side of the positive threshold while the ones w.r.t. the negative class are on the left-hand side of the negative threshold. From Fig. A1 and Fig. A2, we can see that the proposed loss works in this direction, i.e., the attempt pushing all the confidences w.r.t. the positive (negative) class to the right-hand (left-hand) side of the positive (negative) threshold.

To comprehensively evaluate the proposed loss function, we conduct the experiments on various out-of-distribution (OOD) datasets, including ImageNet-C (corrupted ImageNet) [8]. Specifically, we evaluate the trustworthiness predictor trained on ImageNet on the sets of defocus blur, glass blur, motion blur, and zoom blur at the highest level of severity (i.e., the most challenging setting). The results with setting ⟨ViT, ViT⟩ are reported in Table A1. The results are consistent with the ones on the stylized ImageNet and the adversarial ImageNet.

---

[6] `https://github.com/valeoai/ConfidNet`

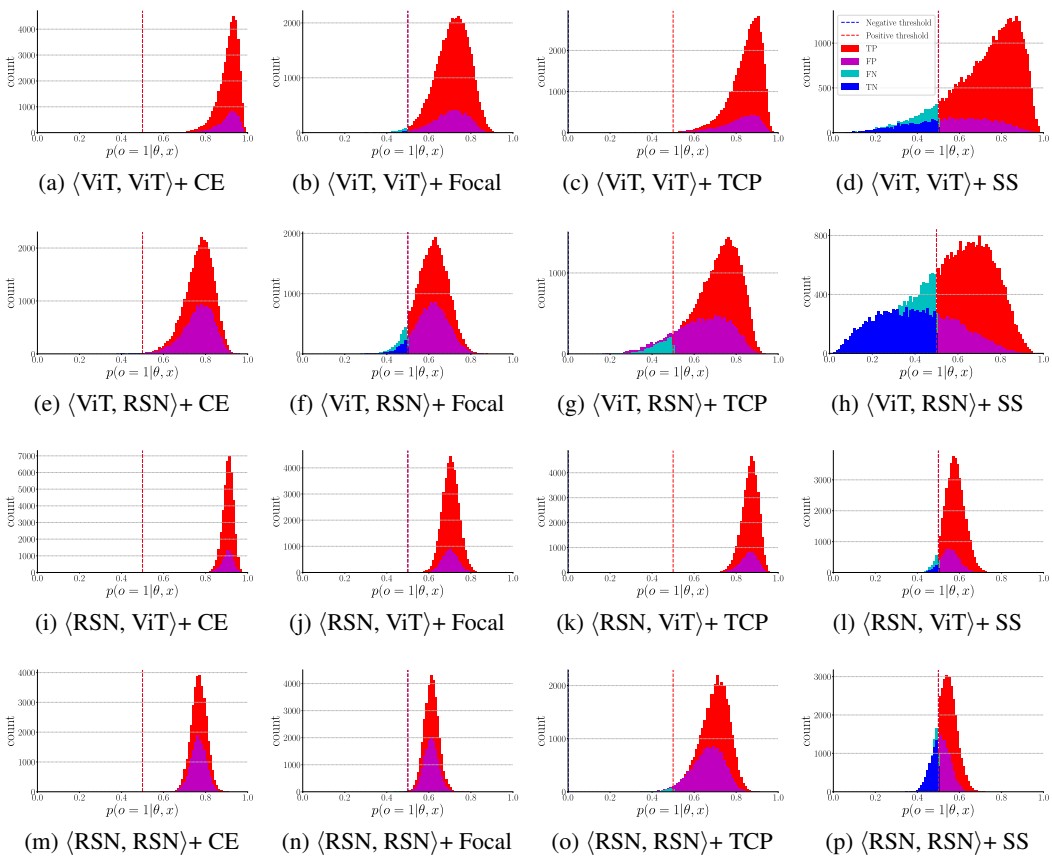

Figure A1: Histograms of trustworthiness confidences w.r.t. all the loss functions on the ImageNet validation set. The oracles that are used to generate the confidences are the ones used in Table 1.

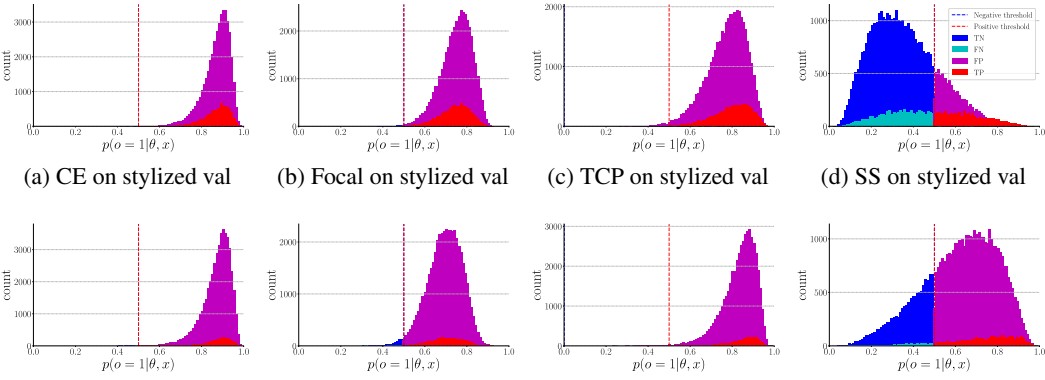

Figure A2: Histograms of trustworthiness confidences w.r.t. all the loss functions on the stylized ImageNet validation set (stylized val) and the adversarial ImageNet validation set (adversarial val). ⟨ViT, ViT⟩ is used in the experiment and the domains of the two validation sets are different from the one of the training set that is used for training the oracle. The histograms correspond to the oracles used in Table 3.

Note that trust score [10] may not be feasible to apply to real-world large-scale datasets like ImageNet. The trust score method needs to hold a tensor of size num_sample × dim_feature to initialize KD trees. The tensor would be small as the trust score method is evaluated on small-scale datasets, e.g.,

Table A1: Performance on ImageNet-C validation set at the highest level of severity [8]. $\langle$ViT, ViT$\rangle$ is used in the experiment and the domains of the two validation sets are different from the one of the training set that is used for training the oracle.

| Set | Loss | Acc↑ | FPR-95%-TPR↓ | AUPR-Error↑ | AUPR-Success↑ | AUC↑ | TPR↑ | TNR↑ |
|---|---|---|---|---|---|---|---|---|
| Defocus blur | CE | 31.83 | 94.46 | 68.56 | 31.47 | 50.13 | 99.15 | 1.07 |
| | Focal [9] | 31.83 | 94.98 | 66.87 | 33.24 | 51.28 | 96.70 | 3.26 |
| | TCP [2] | 31.83 | 93.50 | 64.67 | 36.05 | 54.27 | 96.71 | 4.35 |
| | SS | 31.83 | 90.18 | 57.95 | 48.80 | 64.34 | 77.79 | 37.29 |
| Glass blur | CE | 28.76 | 94.68 | 71.63 | 28.42 | 50.31 | 99.45 | 0.52 |
| | Focal [9] | 28.76 | 92.00 | 67.25 | 33.34 | 56.57 | 96.42 | 6.00 |
| | TCP [2] | 28.76 | 91.24 | 65.64 | 35.72 | 58.78 | 96.95 | 5.50 |
| | SS | 28.76 | 85.03 | 58.58 | 52.55 | 70.18 | 62.68 | 66.25 |
| Motion blur | CE | 43.24 | 94.53 | 56.32 | 43.69 | 50.76 | 99.75 | 0.31 |
| | Focal [9] | 43.24 | 93.67 | 53.83 | 46.47 | 54.02 | 98.88 | 1.55 |
| | TCP [2] | 43.24 | 92.29 | 50.40 | 51.20 | 58.36 | 99.82 | 0.31 |
| | SS | 43.24 | 86.47 | 43.91 | 65.17 | 69.52 | 63.86 | 64.49 |
| Zoom blur | CE | 39.68 | 93.28 | 56.91 | 43.45 | 54.90 | 99.71 | 0.42 |
| | Focal [9] | 39.68 | 92.54 | 54.93 | 46.18 | 57.19 | 98.85 | 1.63 |
| | TCP [2] | 39.68 | 89.47 | 51.79 | 51.06 | 62.47 | 99.64 | 0.93 |
| | SS | 39.68 | 86.53 | 46.74 | 63.84 | 70.94 | 71.58 | 57.38 |

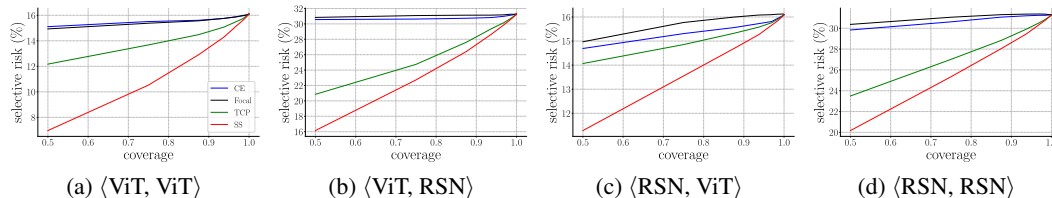

(a) $\langle$ViT, ViT$\rangle$      (b) $\langle$ViT, RSN$\rangle$      (c) $\langle$RSN, ViT$\rangle$      (d) $\langle$RSN, RSN$\rangle$

Figure A3: Curves of risk vs. coverage. Selective risk represents the percentage of errors in the remaining validation set for a given coverage. The curves correspond to the oracles used in Table 1.

$50000 \times 512$ on CIFAR. When evaluating on ImageNet, the size of the tensor would be 1.2 million $\times$ 768 (2048) using ViT (ResNet).

# E    Selective Risk Analysis

Following [11, 2], we present the risk-coverage curves that are generated with different combinations in Fig. A3. As can be seen in the figure, the proposed loss can work with different oracles or classifiers to significantly reduce the error.

# F    Linear Function vs. Signed Distance

Although the signed distance $z$, i.e., $z = \frac{\boldsymbol{w}^\top \boldsymbol{x}^{out} + b}{\|\boldsymbol{w}\|}$, leads to a geometric interpretation as shown in Fig. 2a, the main-stream models [12, 13, 7] use $z = \boldsymbol{w}^\top \boldsymbol{x}^{out} + b$. Therefore, we provide the corresponding comparative results in Table A2, which are generated by the proposed loss taking the output of the linear function as input. In this analysis, the combination $\langle$ViT, ViT $\rangle$ is used and $\alpha^+ = 1, \alpha^- = 3$.

As shown in Table A2, both $z = \frac{\boldsymbol{w}^\top \boldsymbol{x}^{out} + b}{\|\boldsymbol{w}\|}$ and $z = \boldsymbol{w}^\top \boldsymbol{x}^{out} + b$ yield similar performance on metrics FPR-95%-TPR, AUPR-Error, AUPR-Success, and AUC. On the other hand, TPR and TNR are moderately different between $z = \frac{\boldsymbol{w}^\top \boldsymbol{x}^{out} + b}{\|\boldsymbol{w}\|}$ and $z = \boldsymbol{w}^\top \boldsymbol{x}^{out} + b$, when $\alpha^+$ and $\alpha^-$ are fixed. This implies that TPR and TNR are sensitive to $\|\boldsymbol{w}\|$.

Table A2: Performance comparison between $z = \frac{w^\top x^{out}+b}{\|w\|}$ and $z = w^\top x^{out} + b$.

| ⟨O, C⟩+ Loss | $z$ | Acc↑ | FPR-95%-TPR↓ | AUPR-Error↑ | AUPR-Success↑ | AUC↑ | TPR↑ | TNR↑ |
|---|---|---|---|---|---|---|---|---|
| ⟨ViT, ViT ⟩+ SS | $\frac{w^\top x^{out}+b}{\|w\|}$ | 83.90 | 80.89 | 10.31 | 92.90 | 73.31 | 88.44 | 35.64 |
| | $w^\top x^{out}+b$ | 83.90 | 80.42 | 10.36 | 92.78 | 72.93 | 92.40 | 27.01 |
| ⟨ViT, RSN⟩+ SS | $\frac{w^\top x^{out}+b}{\|w\|}$ | 68.72 | 78.09 | 20.94 | 85.30 | 74.20 | 67.96 | 67.80 |
| | $w^\top x^{out}+b$ | 68.72 | 76.65 | 20.75 | 85.82 | 75.00 | 73.54 | 62.66 |
| ⟨RSN, ViT⟩+ SS | $\frac{w^\top x^{out}+b}{\|w\|}$ | 83.90 | 88.63 | 11.75 | 89.87 | 64.11 | 95.41 | 10.48 |
| | $w^\top x^{out}+b$ | 83.90 | 88.39 | 11.65 | 90.06 | 64.56 | 97.95 | 5.45 |
| ⟨RSN, RSN⟩+ SS | $\frac{w^\top x^{out}+b}{\|w\|}$ | 68.72 | 85.60 | 22.50 | 81.92 | 68.08 | 80.44 | 40.85 |
| | $w^\top x^{out}+b$ | 68.72 | 85.59 | 22.42 | 82.02 | 68.33 | 80.03 | 41.82 |

Table A3: Separability of distributions w.r.t. correct predictions and incorrect predictions.

| ⟨O, C⟩ | Loss | $\bar{d}_{KL}$ ↑ | $d_B$ ↑ |
|---|---|---|---|
| ⟨ViT, ViT ⟩ | CE | 0.5139 | 0.0017 |
| | Focal [9] | 0.5212 | 0.0026 |
| | TCP [2] | 0.7473 | 0.0297 |
| | SS | 1.2684 | 0.0947 |
| ⟨ViT, RSN⟩ | CE | 0.5131 | 0.0016 |
| | Focal [9] | 0.5017 | 0.0002 |
| | TCP [2] | 0.9602 | 0.0559 |
| | SS | 1.3525 | 0.1065 |
| ⟨RSN, ViT⟩ | CE | 0.5267 | 0.0033 |
| | Focal [9] | 0.5121 | 0.0015 |
| | TCP [2] | 0.5528 | 0.0066 |
| | SS | 0.7706 | 0.0337 |
| ⟨RSN, RSN⟩ | CE | 0.5077 | 0.0010 |
| | Focal [9] | 0.5046 | 0.0006 |
| | TCP [2] | 0.7132 | 0.0265 |
| | SS | 0.9539 | 0.0565 |

# G   Separability between Distributions of Correct Predictions and Incorrect Predictions

We assess the separability between the distributions of correct predictions and incorrect predictions from a probabilistic perspective. There are two common tools to achieve the goal, i.e., Kullback–Leibler (KL) divergence [14] and Bhattacharyya distance [15]. KL divergence is used to measure the difference between two distributions [16, 17], while Bhattacharyya distance is used to measure the similarity of two probability distributions. Given the distribution of correct predictions $\mathcal{N}_1(\mu_1, \sigma_1^2)$ and the distribution of correct predictions $\mathcal{N}_2(\mu_2, \sigma_2^2)$, we use the averaged KL divergence, i.e., $\bar{d}_{KL}(\mathcal{N}_1, \mathcal{N}_2) = (d_{KL}(\mathcal{N}_1, \mathcal{N}_2) + d_{KL}(\mathcal{N}_2, \mathcal{N}_1))/2$, where $d_{KL}(\mathcal{N}_1, \mathcal{N}_2) = \log \frac{\sigma_2}{\sigma_1} + \frac{\sigma_1^2 + (\mu_1 - \mu_2)^2}{2\sigma_2^2} - \frac{1}{2}$ is not symmetrical. On the other hand, Bhattacharyya distance is defined as $d_B(\mathcal{N}_1, \mathcal{N}_2) = \frac{1}{4} \ln \left( \frac{1}{4} \left( \frac{\sigma_1^2}{\sigma_2^2} + \frac{\sigma_2^2}{\sigma_1^2} + 2 \right) \right) + \frac{1}{4} \left( \frac{(\mu_1 - \mu_2)^2}{\sigma_1^2 + \sigma_2^2} \right)$. A larger $\bar{d}_{KL}$ or $d_B$ indicates that the two distributions are further away from each other.

The separabilities are reported in Table A3. We can see that the proposed loss leads to larger separability than the other three loss functions. This implies that the proposed loss is more effective to differentiate incorrect predictions from correct predictions, or vice versa.

Table A4: Effects of the re-weighting strategy with various loss functions. *CB* stands for class-balanced.

| ⟨O, C⟩ | Loss | Acc↑ | FPR-95%-TPR↓ | AUPR-Error↑ | AUPR-Success↑ | AUC↑ | TPR↑ | TNR↑ |
|---|---|---|---|---|---|---|---|---|
| ⟨ViT, ViT⟩ | CB CE [3] | 83.90 | 85.03 | 11.77 | 89.48 | 65.56 | 99.81 | 0.91 |
| | CB Focal [3] | 83.90 | 82.34 | 10.91 | 91.25 | 69.52 | 99.36 | 2.80 |
| | SS | 83.90 | 80.89 | 10.31 | 92.90 | 73.31 | 88.44 | 35.64 |
| | CB SS | 83.90 | 80.98 | 10.36 | 92.68 | 73.07 | 97.30 | 11.83 |
| ⟨ViT, RSN⟩ | CB CE [3] | 68.72 | 79.30 | 21.99 | 82.25 | 70.74 | 96.32 | 17.10 |
| | CB Focal [3] | 68.72 | 76.89 | 21.10 | 84.59 | 73.75 | 95.71 | 21.15 |
| | SS | 68.72 | 78.09 | 20.94 | 85.30 | 74.20 | 67.96 | 67.80 |
| | CB SS | 68.72 | 75.49 | 20.58 | 86.28 | 75.78 | 89.30 | 39.74 |
| ⟨RSN, ViT⟩ | CB CE [3] | 83.90 | 90.71 | 12.94 | 87.76 | 59.31 | 100.00 | 0.00 |
| | CB Focal [3] | 83.90 | 89.25 | 12.38 | 88.71 | 61.35 | 100.00 | 0.00 |
| | SS | 83.90 | 88.63 | 11.75 | 89.87 | 64.11 | 95.41 | 10.48 |
| | CB SS | 68.72 | 87.28 | 11.51 | 90.34 | 65.34 | 98.28 | 5.23 |
| ⟨RSN, RSN⟩ | CB CE [3] | 68.72 | 86.48 | 23.35 | 79.97 | 65.63 | 99.87 | 0.58 |
| | CB Focal [3] | 68.72 | 85.63 | 22.75 | 81.16 | 67.48 | 99.47 | 2.01 |
| | SS | 68.72 | 85.60 | 22.50 | 81.92 | 68.08 | 80.44 | 40.85 |
| | CB SS | 68.72 | 85.55 | 22.28 | 82.39 | 68.84 | 81.93 | 40.16 |

Table A5: Effects of up-weighting the losses. $w^+$ and $w^-$ are denoted as the weights for the cross entropy losses w.r.t. true samples and negative samples, respectively.

| Loss | $(w^+, w^-)$ | FPR-95%-TPR↓ | AUPR-Error↑ | AUPR-Success↑ | AUC↑ | TPR↑ | TNR↑ |
|---|---|---|---|---|---|---|---|
| CE | (1, 1) | 93.01 | 15.80 | 84.25 | 51.62 | 99.99 | 0.02 |
| CE | (1, 5) | 93.54 | 15.57 | 84.47 | 52.09 | 97.86 | 2.98 |
| CE | (1, 10) | 93.49 | 15.66 | 84.36 | 51.48 | 46.04 | 55.52 |
| CE | (1, 15) | 94.80 | 16.09 | 83.93 | 50.77 | 5.06 | 94.52 |
| CE | (1, 20) | 93.90 | 15.98 | 84.08 | 51.53 | 0.26 | 99.52 |
| SS | (1, 1) | 80.48 | 10.26 | 93.01 | 73.68 | 87.52 | 38.27 |

# H  Connection to Class-Balanced Loss

The class-balanced loss [3] is actually to apply the re-weighting strategy to a conventional loss function, e.g., the cross entropy loss and the focal loss. To re-weight the losses, it presumes to know the number of sample w.r.t. each class before training, i.e., $n^+$ (the number of correct predictions) and $n^-$ (the number of incorrect predictions). Following [3], we use the hyperparameters $\beta = 0.999$ and $\gamma = 0.5$ for the class-balanced loss. The class-balanced loss can be also applied to the proposed loss.

We report the performances of the class-balanced cross entropy loss, the class-balanced focal loss, the proposed loss and its class-balanced variant in Table A4. The class-balanced cross entropy loss and focal loss achieve better performance on most of metrics than the cross entropy loss and focal loss. Consistently, the proposed loss and its class-balanced variant outperform the class-balanced cross entropy loss and focal loss on metrics AUPR-Success, AUC, and TNR. On the other hand, the class-balanced steep slope loss does not comprehensively outperform the proposed steep slope loss. The improvement gain from the re-weighting strategy used in the imbalanced classification task is limited. This may result from the difference between the imbalanced classification and trustworthiness prediction. In other words, the imbalanced classification is aware of the visual concepts, whereas the trustworthiness is invariant to visual concepts.

Up-weighting (down-weighting) the losses w.r.t. negative (positive) samples is a naive strategy in the imbalanced classification. It is interesting to see if this simple strategy is able to address the problem of predicting trustworthiness. Let $w^+$ and $w^-$ be the weights for the cross entropy losses w.r.t. positive samples and negative samples, respectively. The experimental results are in Table A5. As we can see, up-weighting $w^-$ with various values does not achieve desired performance. In contrast, applying the class-balanced strategy for re-weighting yields much better results than the

Table A6: Numerical stability of gradients. $\langle \text{ViT, ViT} \rangle$ is used for the analysis.

| Magnitude | CE | Focal | TCP | SS |
|---|---|---|---|---|
| $\|\frac{\partial \ell}{\partial x^{out}}\|$ | 0.0068 | 0.0041 | 0.0039 | 0.0149 |

naive up-weighting strategy. Moreover, the proposed loss achieves better performance than the naive up-weighting strategy.

## I  Analysis of Gradients

It is interesting to know whether there are vanishing gradient issues or numerical stability issues due to the exponent in the proposed loss function. We compute the averaged $\|\frac{\partial \ell}{\partial x^{out}}\|$, where $x^{out} \in \mathbf{R}^k$ is the dimension of the output feature of the oracle backbone. For the case of using ViT as backbone, $k = 768$. The results with setting $\langle \text{ViT, ViT} \rangle$ are reported in Table A6. Although the proposed loss function uses the exponential function, it only involves a small range (e.g., $[\exp(-\alpha^+), \exp(\alpha^+)]$ or $[\exp(-\alpha^-), \exp(\alpha^-)]$) in the exponential function. In this range, the gradients are less likely to change dramatically.

## J  Inference

Once the training process for the trustworthiness predictor is done, the inference by the trustworthiness predictor is efficient. Specifically, the inference time used for classification is 1.64 milliseconds per image, while the inference time used for predicting trustworthiness is 1.53 milliseconds per image. Predicting trustworthiness is slightly faster than predicting labels. This is because the classification is a 1000-way prediction while predicting trustworthiness is a 1-way prediction (I.e., the output is a scalar). As the trustworthiness predictor and classifier are separate, it is possible to predict trustworthiness and labels in parallel to further improve efficiency in practice.