# OpenReview forum: "Learning to Predict Trustworthiness with Steep Slope Loss"
_NeurIPS.cc/2021/Conference — NeurIPS 2021 Poster_

### Official Review · Reviewer_m14X · 2021-07-15

**Rating:** 7
**Confidence:** 3

**Summary:**

The paper proposes the steep slope loss for learning trustworthiness under a supervised training setup. A pre-trained neural network is further trained on the steep slope loss to learn trustworthiness as a binary classification problem. The approach is evaluated on ImageNet for a ResNet and a ViT model. In addition, evaluations on MNIST and CIFAR-10 are provided for comparisons with the related work. In all experiments, the proposed approach shows good performance.


**Limitations And Societal Impact:**

-

**Main Review:**

Paper strengths:

+ The paper addresses an open problem where there is not a standard solution. It's well-written and mostly easy to follow.

+ The steep slope loss is well-motivated. The visual illustration of the loss in Fig. 2 helps to immediately follow how it works.

+ The evaluation relies on the standard metrics for the presented problem. Also, recent approaches are included in the evaluations.  It's clear that the proposed loss works well for most of the metrics.

+ The paper is the first one to perform a large-scale evaluation for trustworthiness estimation.

+ Unseen domain evaluation: the proposed approach makes use of trustworthiness estimation for identifying wrong predictions in a second domain. This is a novel application for trustworthiness that has not been presenter before.


Paper weaknesses:

- The small-scale evaluation is important for the paper because one can compare it with the prior work. They should have been part of the main paper.

- It's not clear whether the trustworthiness estimator (oracle) and the classifier are two separate models. If so, it's important to discuss the inference time of the presented algorithms. The trustworthiness estimator would probably double the inference.

- Comparison with Trust Scores (To trust or not to trust a classifier, NIPS, 2018): The trust scores is a related approach for trustworthiness estimation. It would be supportive for the paper to compare with this approach.


Improvements:

- line 96: m is not defined.

- Eq. 4 / Eq. 5: what is the first minus?


Overall:

The paper presents an interesting approach for trustworthiness estimation. The proposed loss function learns when a deep neural network makes untrustworthy predictions. The main innovation of the paper is the evaluation of ImageNet using two complex models. The proposed study brings trustworthiness estimation closer to real-world problems. It's the first one to consider ImageNet with recent network architectures such as the transformer.


After rebuttal comments:

The rebuttal addressed my points. In addition, the major concerns on the theoretical study, the difference between the proposed method and negative sample up-weighting, and Imagenet-Corrupted evaluation are discussed in the rebuttal. The proposed approach is the standard evaluation protocol for trustworthiness estimation. The approach does not have to be evaluated for out-of-distribution detection. The current protocol is completely fine and similar to the prior work. For these reasons, I retain my positive evaluation since the paper has enough merit to be accepted.


**Time Spent Reviewing:**

2-3

---

> ### Author Response · Authors · 2021-08-10
> **Thank you for the insightful suggestions**
>
> We thank the reviewer for the in-depth feedback! We are encouraged that the reviewer finds the work "well-motivated", "well written", and "easy to follow". We are pleased that the reviewer recognizes that the paper "addresses an open problem where there is not a standard solution", "brings trustworthiness estimation closer to real-world problems" as "the first one to perform a large-scale evaluation for trustworthiness estimation", and "unseen domain evaluation has not been presented before". We address the questions/comments below and will incorporate all feedback in the final version.
>
> **m14X-Q1**: The small-scale evaluation is important for the paper because one can compare it with the prior work. They should have been part of the main paper.
>
> **Response**: We thank the reviewer for the suggestion. We will include Table A2 in the main paper in the revision.
>
> **m14X-Q2**: It's not clear whether the trustworthiness estimator (oracle) and the classifier are two separate models. If so, it's important to discuss the inference time of the presented algorithms. The trustworthiness estimator would probably double the inference.
>
> **Response**: We thank the reviewer for this constructive comment. We will add the discussion of inference time to the revision.
>
> We follow the design of TCP [r1], where the confidence model plays the role of the trustworthiness predictor alongside the classification model. The two models are separate but part of their weights are set to be the same (see the TCP [official implementation]( https://github.com/valeoai/ConfidNet/blob/4b2607fe93a77f50cb95eb8b854b21591ef8f478/confidnet/models/vgg16_selfconfid_cloning.py#L18-L19)).
>
> Once the training process for the trustworthiness predictor is done, the inference by the trustworthiness predictor is efficient. Specifically, the inference time used for classification is 1.64 milliseconds per image, while the inference time used for predicting trustworthiness is 1.53 milliseconds per image. Predicting trustworthiness is slightly faster than predicting labels. This is because the classification is a 1000-way prediction while predicting trustworthiness is a 1-way prediction (i.e., the output is a scalar). As the trustworthiness predictor and classifier are separate, it is possible to predict trustworthiness and labels in parallel to further improve efficiency in practice.
>
> **m14X-Q3**: Comparison with Trust Scores (To trust or not to trust a classifier, NIPS 2018): The trust scores is a related approach for trustworthiness estimation. It would be supportive for the paper to compare with this approach.
>
> **Response**: Thanks for bringing this to the discussion. Trust score [r2] may not be a good fit to tackle real-world large-scale datasets like ImageNet. The trust score method needs to hold a tensor of size num_sample $\times$ dim_feature to initialize a KD tree. The tensor would be small as the trust score method is evaluated on small-scale datasets, e.g., 50000 $\times$ 512 on CIFAR. When evaluated on ImageNet, the size of the tensor would be 1.2 million $\times$ 768 (2048) using ViT (ResNet). We use the official implementation [r2] based on sklearn library. It turns out that the program for computing the trust score always exits abnormally.
>
> **m14X-C4**: Improvements
> - line 96: m is not defined.
>
> - Eq. 4 / Eq. 5: what is the first minus?
>
>
> **Response**: We thank the reviewer for pointing out the missing definition of $m$, which is the number of classes. We will revise the text accordingly in the revision.
> In Eq. (4) and (5), the first minus is part of the negative log likelihood.

---

> ### Author Response · Authors · 2021-08-26
> **Thanks for updating the review with "after rebuttal comments"**
>
> We thank the reviewer for the encouraging and positive comments about our responses and the recognition of the merit of the paper. We also thank the reviewer for acknowledging our approach and protocol used.

---

### Official Review · Reviewer_R52P · 2021-07-17

**Rating:** 6
**Confidence:** 2

**Summary:**

The authors find that past trustworthiness predicting methods are prone to overfitting to the training set where correct predictions are dominant. The label imbalance makes the trustworthiness predictors over-confident to even incorrect predictions. To solve the problem, this work proposes a new training loss named steep slope loss, which aims at improving the generalizability of trustworthiness predictors. The intuition behind this is to emphasize the negative class during training. The experiments are conducted in ImageNet. The results show the shortcomings of past methods with cross-entropy loss, focal loss, and TCP loss and then demonstrate the effectiveness of the proposed method.

**Limitations And Societal Impact:**

Yes, the authors have adequately addressed the limitations.

**Main Review:**

*Strengths
- The paper reveals the shortcomings of past works in large-scale datasets like ImageNet, which is an interesting problem.
- The paper proposes an effective method to improve the generalizability for trustworthiness predictors.

*Weaknesses
- The paper lacks a theoretical study about why the proposed loss is better than the previous methods.
- It is not clear to the reviewer the empirical and theoretical difference between the proposed method and simply up-weighting the negative samples in trustworthiness learning.

**Time Spent Reviewing:**

20

---

> ### Author Response · Authors · 2021-08-10
> **Thank you for the constructive comments**
>
> We thank the reviewer for the valuable feedback! We are encouraged that the reviewer finds our work "reveals the shortcomings of past works in large-scale datasets" and recognizes our contribution as "an effective method to improve the generalizability for trustworthiness predictors". We address the questions/comments below and will incorporate all feedback in the final version.
>
> **R52P-Q1**: The paper lacks a theoretical study about why the proposed loss is better than the previous methods.
>
> **Response**: We provide a theoretical study that the proposed loss has the generalization bound in Theorem 3.1. Meanwhile, the cross entropy loss, focal loss, and TCP loss are not capped in a certain range so that they are not feasible to derive similar generalization bounds. Specifically, as shown in Theorem 3.1 and Figure 2b, the proposed loss is capped in some range, i.e., $\ell_{SS}\in [0, \ell_{SS}^{max}]$. In contrast, $\ell_{CE}, \ell_{Focal}, \ell_{TCP} \in [0, +\infty)$. As shown in the proof (Appendix A), these loss functions do not fit under Hoeffding's inequality to derive the generalization bounds. In summary, the proposed loss function has the generalization bound whereas the generalization bounds of the other three loss functions are unclear. We will revise the text to make this point clearer.
>
> **R52P-Q2**: It is not clear to the reviewer the empirical and theoretical difference between the proposed method and simply up-weighting the negative samples in trustworthiness learning.
>
> **Response**: We thank the reviewer for the suggestion. Up-weighting the losses w.r.t. negative samples is a naive strategy in the imbalanced classification. We provide the experimental results with the cross entropy loss with setting <ViT, ViT> on ImageNet below. Moreover, the paper includes (see line 324 -- 328) the experimental results of a state-of-the-art loss for the imbalanced classification, (i.e., class-balanced loss [r7]), that aims to re-weight the true/negative samples smartly. Let $w^{+}$ and $w^{-}$ be the weights for the cross entropy losses w.r.t. positive samples and negative samples, respectively.
>
> Instead, the proposed loss uses $\alpha^{+}$ and $\alpha^{-}$ to manage the degree in separating positive samples from negative samples. Also, as discussed in Appendix I, the proposed loss is able to work with any re-weighting strategies.
>
> As we can see, up-weighting $w^{-}$ with various values does not achieve desired performance. In contrast, as shown in Table A5, applying the class-balanced strategy for re-weighting yields much better results than the naive up-weighting strategy. Moreover, the proposed loss achieves better performance than the naive up-weighting strategy. We will add the discussion to the revision.
>
> | Loss | ($w^{+}$, $w^{-}$)| FPR-95%-TPR$\downarrow$ | AUPR-Error$\uparrow$ | AUPR-Success$\uparrow$ | AUC$\uparrow$ | TPR$\uparrow$ | TNR$\uparrow$ |
> | :---        |    :----:   |  :----: | :----: | :----: | :----: | :----: | :----: |
> | CE | (1, 1) | 93.01 | 15.80 | 84.25 | 51.62 | 99.99 | 0.02 |
> | CE | (1, 5) | 93.54 | 15.57 | 84.47 | 52.09 | 97.86 | 2.98 |
> | CE | (1, 10) | 93.49 | 15.66 | 84.36 | 51.48 | 46.04 | 55.52 |
> | CE | (1, 15) | 94.80 | 16.09 | 83.93 | 50.77 | 5.06 | 94.52 |
> | CE | (1, 20) | 93.90 | 15.98 | 84.08 | 51.53 | 0.26 | 99.52 |
> | SS | (1, 1) | 80.48 | 10.26 | 93.01 | 73.68 | 87.52 | 38.27 |
>
>
> **References**:
>
> [r7] Yin Cui, Menglin Jia, Tsung-Yi Lin, Yang Song, and Serge Belongie. Class-balanced loss
> based on effective number of samples. CVPR 2019.

---

> > ### Comment · Reviewer_R52P · 2021-08-19
> > **comments to the authors response.**
> >
> > Thanks to the authors for the detailed responses. I am OK with it.

---

> > > ### Author Response · Authors · 2021-08-26
> > > **Thanks for the confirmation**
> > >
> > > We thank the reviewer for reading and acknowledging our responses. We really appreciate the comments and discussions to help improve our work.

---

### Official Review · Reviewer_8gNS · 2021-07-19

**Rating:** 8
**Confidence:** 4

**Summary:**

The paper proposes Steep Slope loss, a loss function that is designed to improve performance in the binary classification task of prediction trustworthiness. The paper reports that common loss functions used in this domain such as cross-entropy loss, focal loss, and true classification probability loss (TCP), all suffer from an issue where correct predictions tend to dominate over incorrect predictions, due to issues of class imbalance and evaluations on simple datasets where classifiers have achieved very high accuracy.

Steep Slope loss solves this issue by controlling the slope parameters on both the negative and positive classification thresholds with two hyperparameters that control the derivative and gradient updates for datapoints that are correctly classified and incorrectly classified separately. These two controlled slopes are a function of the signed distance of the output of the classifier from the hyperplane of classification.

The paper evaluates this loss on the MNIST, CIFAR, and Imagenet datasets and shows empirical improvements on multiple binary classification metrics such as AUPRC, AUROC, and FPR95, and importantly, SS loss shows marked improvements in the TNR, therefore being more robust to detecting incorrectly classified points.

**Ethical Concerns:**

Not to my knowledge.

**Limitations And Societal Impact:**

The authors have responded N/A to this prompt in the checklist. However, I would advise the authors to add a small discussion about the importance of trading off between TPR and TNR for critical applications such as in medicine, where the robustness of their method to TNR would make a strong statement.

**Main Review:**

### Strengths of the Work -

The paper is well written, motivates the problem of trustworthiness classification well, and derives a well-argued formulation for Steep Slope loss. The empirical results are quite strong, and the datasets studied are comprehensive as well.

The formulation for SS loss seems like it will be quite beneficial for many binary classification tasks as well, especially in order to improve model calibration on in-domain data. This is a promising loss function to add to the family for more than trustworthiness applications as well.

The comparisons to class-imbalance loss and clear reasons for why SS loss is a better choice also adds to the strength of the argument in a domain where the "positive class" and "negative class" statistics are unknown and depend on the accuracy of the classifier being studied.

Showing the robustness of this loss to unseen domains such as the stylized and adversarial Imagenet test sets also strengthens the method, and the TNR numbers in Table 2 are quite impressive compared to the baselines.

### Comments and Suggestions -

The paper is a good contender for acceptance in its current form. The appendices are comprehensive, and the following points, if addressed, would strengthen my opinion of acceptance further.

1. It would be interesting to see a numerical analysis of gradients of the Steep Slope loss function, whether there are vanishing gradient issues, or numerical stability issues due to the exponent.
2. Another potential test dataset to evaluate the robustness of SS loss could be the Imagenet-Corrupted datasets, defined in [1]. Trustworthiness as a function of dataset shift can be an interesting property to map, to strengthen the adoption of SS loss to domains where there can be covariate shift from the training regime to an unknown test regime.
3. Why are the TNR results so sensitive to the normalization by $||w||$? I would be curious to know if the authors have a hypothesis for why it is better to normalize the affine outputs with a norm of the weights?
4. The paper mentions that all the oracles were trained with the same hyperparameters (LR, weight decay, momentum etc). This doesn't make much sense to me, as each of these loss functions have very different training behaviours and gradient updates, and the number of steps required for convergence would be different. Further, since the loss  landscape for these loss functions are different, the optimal hyperparameter space would be different for each of the loss functions. It might be that the SS loss is in a good hparam space for the current hparams, but the other losses are not.
5. Could the authors clarify how the picked the values of $\alpha^+$ and $\alpha^-$ for different datasets, or point me to the relevant section in the paper in case I missed it?
6. From the histogram plots, it seems like that a lot of True Positive Examples get misclassified, with the tradeoff that more True Negative Examples are accurately classified. It would be good for the authors to discuss the pros and cons of this tradeoff, especially in real-world applications of trustworthiness (medical applications etc), which can strengthen arguments for their method.
7. A further comparison to non-loss functions method in the trustworthiness sphere could be good, such as the trust score [2].

Nitpicks -

Line 45: "state-of-the-art"

Line 84: should be $f_\theta'$ instead of $f_\theta$

Line 136: "hyperplane"

Line 137: $x_{\text{out}}$ instead of $x_{\text{in}}$


### Originality of the work

To my knowledge, the formulation of the SS loss is novel, and the parametrization of positive and negative slopes to aid with the class imbalance present in trustworthiness classification has not been done before. There is rich literature in the field of proper scoring rules and binary classification for parametrized binary loss functions [1], but I have not seen any research in the field of trustworthiness that makes use of these parametrized loss functions. This work has definite novelty in scaling the methods to Imagenet and ViT, whereas previous work in this field focused on MNIST and CIFAR, which are much easier tasks with higher classification accuracy.

### Quality and clarity

The paper is well-written, hyperparameter tuning and ablation experiments are comprehensive, and the proof for the generalization bound seems correct to me. The authors have picked valid metrics to display the pros and cons of their method, though a further discussion on balancing the tradeoff between TPR and TNR would be good.

### Significance

In my opinion, the empirical results are impressive and push the envelope for binary classification in the trustworthiness sphere. The implementation of the loss function is also straightforward in a standard deep learning package, and can be plugged into existing pipelines with a one-line change. The authors also perform experiments on a large scale dataset such as Imagenet and report state-of-the-art numbers for their method and some baselines. The risk analysis plots are also an additional way of testing the robustness of these methods, and further results on synthetic and adversarial versions of the test dataset should be considered the norm for future analysis in this space.



**Time Spent Reviewing:**

4

---

> ### Author Response · Authors · 2021-08-10
> **Thank you for the constructive, detailed, and insightful suggestions**
>
> We thank the reviewer for the in-depth feedback! We are encouraged that the reviewer finds our work "novel" and "well written" with "well-argued formulation". We are glad that the reviewer identifies that "the empirical results are impressive", "the datasets studied are comprehensive", and "this is a promising loss function to add to the family for more than trustworthiness applications as well". We address the questions/comments below and will incorporate all feedback in the final version.
>
> **8gNS-Q1**: It would be interesting to see a numerical analysis of gradients of the Steep Slope loss function, whether there are vanishing gradient issues or numerical stability issues due to the exponent.
>
> **Response**: We thank the reviewer for this suggestion on numerical stability. We compute the averaged $\|\|\frac{\partial \ell}{\partial x^{out}}\|\|$, where $x^{out} \in \mathbf{R}^{k}$ is the dimension of the output feature of the oracle backbone. For the case of using ViT as backbone, $k=768$. The results with setting <ViT, ViT> are as follows.
>
> |          | CE | Focal | TCP | SS     |
> | :---        |    :----:   |  :----:  | :----:  | :----:  |
> | $\|\|\frac{\partial \ell}{\partial x^{out}}\|\|$ | 0.0068 | 0.0041 | 0.0039 | 0.0149 |
>
> Although the proposed loss function uses the exponential function, it only involves a small range (e.g., $[\exp(-\alpha^{+}), \exp(\alpha^{+})]$ or $[\exp(-\alpha^{-}), \exp(\alpha^{-})]$) in the exponential function. In this range, the gradients are less likely to change dramatically.
>
> **8gNS-Q2**: Another potential test dataset to evaluate the robustness of SS loss could be the Imagenet-Corrupted datasets, defined in [1].
>
> **Response**: Thanks for recommending this dataset to improve the quality of the work. We assume that the recommended dataset is [r6]. We evaluate the trustworthiness predictor trained on ImageNet on the sets of defocus blur, glass blur, motion blur, and zoom blur at the highest level of severity (i.e., the most challenging setting). The results with setting <ViT, ViT> are as follows. The results are consistent with the ones on the stylized ImageNet (c.f. Table 2 in the paper) and the adversarial ImageNet (c.f. Table 2 in the paper).
>
> | Dataset | Loss | FPR-95%-TPR$\downarrow$ | AUPR-Error$\uparrow$ | AUPR-Success$\uparrow$ | AUC$\uparrow$ | TPR$\uparrow$ | TNR$\uparrow$ |
> | :---        |    :----:   |  :----: | :----: | :----: | :----: | :----: | :----: |
> | Defocus blur |  CE | 94.46 | 68.56 | 31.47 | 50.13 | 99.15 | 1.07 |
> | Defocus blur |  Focal | 94.98  | 66.87 | 33.24 | 51.28 | 96.70 | 3.26 |
> | Defocus blur | TCP |  93.50 | 64.67 | 36.05 | 54.27 | 96.71 | 4.35 |
> | Defocus blur |  SS |  90.18 | 57.95 | 48.80 | 64.34 | 77.79 | 37.29 |
> | Glass blur |  CE | 94.68 | 71.63 | 28.42 | 50.31 | 99.45 | 0.52 |
> | Glass blur |  Focal | 92.00 | 67.25 | 33.34 | 56.57 | 96.42 | 6.00 |
> | Glass blur | TCP |  91.24 | 65.64 | 35.72 | 58.78 | 96.95 | 5.50 |
> | Glass blur |  SS |  85.03 | 58.58 | 52.55 | 70.18 | 62.68 | 66.25 |
> | Motion blur |  CE | 94.53 | 56.32 | 43.69 | 50.76 | 99.75 | 0.31 |
> | Motion blur |  Focal | 93.67 | 53.83 | 46.47 | 54.02 | 98.88 | 1.55 |
> | Motion blur | TCP |  92.29 | 50.40 | 51.20 | 58.36 | 99.82 | 0.31 |
> | Motion blur |  SS |  86.47 | 43.91 | 65.17 | 69.52 | 63.86 | 64.49 |
> | Zoom blur |  CE | 93.28 | 56.91 | 43.45 | 54.90 | 99.71 | 0.42 |
> | Zoom blur |  Focal | 92.54 | 54.93 | 46.18 | 57.19 | 98.85 | 1.63 |
> | Zoom blur | TCP |  89.47 | 51.79 | 51.06 | 62.47 | 99.64 | 0.93 |
> | Zoom blur |  SS |  86.53 | 46.74 | 63.84 | 70.94 | 71.58 | 57.38 |
>
> **8gNS-Q3**: Why are the TNR results so sensitive to the normalization by |w|? I would be curious to know if the authors have a hypothesis for why it is better to normalize the affine outputs with a norm of the weights?
>
> **Response**: The normalization by $\|w\|$ would make $z$ more dispersed in value than the variant without normalization. In other words, the normalization leads to long-tailed distributions while no normalization leads to short-tailed distributions. Given the same threshold, TNR (TPR) is determined by the location of the distribution of negative (positive) examples and the extent of short/long tails. We plot out the histograms of the trustworthiness predictor without normalization. By comparing the histograms to the ones in Figure 3, the histograms generated without normalization are more spread than the ones generated with normalization, which verifies this point. We will add the discussion and the histogram plots to the revision.
>
> **8gNS-Q4**: It might be that the SS loss is in a good hparam space for the current hparams, but the other losses are not.
>
> **Response**: The optimal hyperparameters are searched and determined by using the cross entropy loss and the focal loss, rather than the proposed SS loss. In our preliminary experiments, we find that the hyperparameters work well with the cross entropy loss and the focal loss. Then, we fix them for the experiments with all the loss functions. This is elaborated in Appendix B. We apologize for the confusion and will clarify this in the paper.
>
> **8gNS-Q5**: Could the authors clarify how they picked the values of alpha+ and alpha- for different datasets, or point me to the relevant section in the paper in case I missed it?
>
> **Response**: The derivation that determines $\alpha^{+}$ and $\alpha^{-}$ is introduced in the ablation study (lines 302--303) in the paper. Briefly, it is an alternating process, i.e., freezing one of the hyperparameters and searching for the other one. We apply this strategy on ImageNet, MNIST, and CIFAR (c.f. Appendix F).
>
> **8gNS-Q6**: From the histogram plots, it seems like that a lot of True Positive Examples get misclassified, with the tradeoff that more True Negative Examples are accurately classified. It would be good for the authors to discuss the pros and cons of this tradeoff, especially in real-world applications of trustworthiness (medical applications etc), which can strengthen arguments for their method.
>
> **Response**: We thank the reviewer for this insightful comment and the recognition of this point. We have followed the suggestions and added a paragraph on the societal impact that includes a discussion of the tradeoff in real-world applications (please refer to the response to Q9). We would also like to add that since the goal of trustworthiness predictors is to distinguish positive examples from negative examples or vice versa, better separability between the two distributions as shown in the proposed method (Figure 3 and lines 315 -- 323 in the paper) is key.
>
> **8gNS-Q7**: A further comparison to non-loss functions method in the trustworthiness sphere could be good, such as the trust score [2].
>
> **Response**: We thank the reviewer for the suggestion. Trust score [r2] may not be feasible to apply to real-world large-scale datasets like ImageNet. The trust score method needs to hold a tensor of size num_sample $\times$ dim_feature to initialize a KD tree. The tensor would be small as the trust score method is evaluated on small-scale datasets, e.g., 50000 $\times$ 512 on CIFAR. When evaluating on ImageNet, the size of the tensor would be 1.2 million $\times$ 768 (2048) using ViT (ResNet). We use the official implementation [r2] based on sklearn library. It turns out that the program for computing the trust score always exits abnormally.
>
> **8gNS-Q8**: "Nitpicks"
>
> **Response**: Thanks for the "nitpicks". We will revise the text accordingly in the revision.
>
>
> **8gNS-Q9**: The authors have responded N/A to this prompt in the checklist. However, I would advise the authors to add a small discussion about the importance of trading off between TPR and TNR for critical applications such as in medicine, where the robustness of their method to TNR would make a strong statement.
>
> **Response**: We thank the reviewer for the very helpful suggestion. We will add the following section on societal impact.
>
> Learning high-accuracy models is a long-standing goal in machine learning. Nevertheless, due to the complexity of real-world data, there is still a gap between state-of-the-art classification models and a perfect one. Hence, there is a critical need to understand the trustworthiness of classification models, i.e., finding out correct predictions and incorrect predictions, in order to safely and effectively apply the models in real-world tasks. Models with the proposed loss achieve considerably better performance with various trustworthiness metrics. They also show generalizability with both in-distribution and out-of-distribution images at a large scale. In addition, while existing trustworthiness models focus on a high true positive rate (TPR) and tend to view all incorrect predictions to be trustworthy (i.e., TNR close to 0), fasle positives may lead to high consequent cost in some real-world scenarios such as critical applications in medicine (e.g., a false positive leading to unnecessary and invasive tests or treatment such as biopsies or surgery, or harmful side effects in medicine) and security (e.g., counter terrorism). It is thus of great importance for a trustworthiness model to flexibly trade off between TPR and TNR. To this end, the proposed loss allows the underlying distributions of positive and negative examples to be more separable, enabling a more effective trade-off between them.
>
>
> **References**:
>
> [r6] Dan Hendrycks and Thomas Dietterich. "Benchmarking Neural Network Robustness to Common Corruptions and Perturbations." ICLR 2018.

---

> > ### Comment · Reviewer_8gNS · 2021-08-24
> > **Response to Rebuttal**
> >
> > I thank the reviewers for their detailed and comprehensive response! I believe that the paper is quite strong, especially with these additions to the supplementary/original text, and I thank them for the time and effort to run these experiments to answer my queries. I stand by my original rating, and would definitely advocate for acceptance based on the original paper + rebuttals received.
> >
> > 1. "it only involves a small range (e.g.,  or ) in the exponential function". Thanks, this answers my query about numerical stability, as it seems like the exp function should be roughly linear/stable in this regime.
> >
> > 2. "the normalization leads to long-tailed distributions while no normalization leads to short-tailed distributions", I buy this argument, and it checks out with the empirical results.
> >
> > 3. "We assume that the recommended dataset is [r6]", apologies about that, it seems I forgot to add the citations at the end of my review. This is indeed the correct paper, and the empirical results on corrupted Imagenet are quite strong.
> >
> > 4. "we find that the hyperparameters work well with the cross entropy loss and the focal loss. Then, we fix them for the experiments with all the loss functions." I still don't think this is a good strategy when using loss functions. In my opinion, you shouldn't be porting hyperparameters found on one loss function to another. If anything, SS loss might work even better if tuned independently. Especially if you are advocating to only using SS loss, you would never have to tune on the cross-entropy loss or focal loss experiments, and should tune independently on SS loss, to see how the hparam space looks and how robust it is to different settings.
> >
> > 5. "The trust score method needs to hold a tensor of size num_sample  dim_feature to initialize a KD tree." Ah, that's a good point, it seems this is impossible on Imagenet, thank you for clarifying.
> >
> > I appreciate the discussion between TPR and TNR, and thank the authors for a strong rebuttal.

---

> > > ### Author Response · Authors · 2021-08-26
> > > **Thanks for the encouraging and positive feedback**
> > >
> > > We thank the reviewer for the additional and very helpful discussions about our responses. We really appreciate the reviewer's expertise and time for improving the work.

---

### Official Review · Reviewer_DFoG · 2021-07-22

**Rating:** 5
**Confidence:** 2

**Summary:**

This paper tests methods for training a classifier for predicting trustworthiness-- formulated here as predicting whether a given image will be classified correctly or not by an target classifier -- on large scale datasets and finds that using existing loss functions like uncertainty estimates/focal loss on this binary classification task cannot detect untrustworthy samples, all samples are classified as positive resulting in 100% FPR (0% TNR). They attempt to fix this shortcoming by introducing a steep slope loss which essentially separates features w.r.t. correct and incorrect predictions from each other, and the loss is bounded in its magnitude. The paper extensively tests whether its possible to predict trustworthiness with two oracle/target classifier combinations on Imagenet and its stylized/adversarial test sets.

**Limitations And Societal Impact:**

No substantial negative societal impact that must've been discussed in my view but the section was missing from the draft.


**Main Review:**

Strengths:

2.1) Clear and well-presented: The contributions are clearly stated and well-motivated, acknowledgement is given to the existing body of work to the best of my knowledge. Experiments are performed on large-scale and existing loss functions fail to differentiate between samples. The paper then proceeds to describe a sufficient solution, results are clear and well-presented-- it seems a good step forward.

2.2) Very significant results and interesting transferability: The increase in TNR seems very significant compared to other loss functions, essentially going from near 0 to very significant distinguishability--which says steep slope loss is very useful. The results hold true for stylized and adversarial sets as well with equally significant degrees. The phenomena of ViT being able to detect RSN trustworthy samples far better than the other way around is a very interesting observation, the difference seems stark enough to go beyond simply it having more classification power.

2.3) Reproducible & nice ablations: The code is provided alongside with detailed readme which I wanted to highlight  (I didn't verify the code but it should work well). The ablation with the balanced loss was well-though of and addressed the question I was thinking of.

Weaknesses:

3.1) Formulation unconvincing [Critical]: In my opinion, trustworthiness of a classifier’s output depends on the classifier. Consequently, the input to the trustworthiness predictor should have that output of the target classifier passed (softmax probability/logit etc) alongside the image. This formulation has only the implicit encoding by using supervised samples of a classifier without providing explicit knowledge about the classifier itself seems badly designed (in principle) and I strongly oppose this.

3.2) Unfair comparisons [Critical]: Past works to the best of my knowledge the trustworthiness (typically used for out-of-distribution detection rather than rejecting test samples) is determined by the probability outputs of the target classifiers (a function of the target classifier). The calibrated probability estimates for the same are obtained by using loss functions like cross-entropy (with some calibration) {13} and focal loss [1] -- not training a separate binary classifier with these loss functions as done here, along with assumption of having access to just the image. Recent (parallel) work [2] has benchmarked these methods on large-scale datasets which indicates these scale well to larger datasets. A fair comparison in my opinion should be with (older-- {13},[1]) classifiers using the output of target networks to determine trustworthiness.

Overall:

The paper is well-written and the improvements are strong. The proposed approach seems simple, accurate and interesting. I mainly have the above concerns regarding the framework introduced and the comparisons made. Overall, I am on the fence but will be willing to increase my rating if the weaknesses are addressed. Note that I might have missed important contribution novelty (summary/strengths) or misunderstood claims (weaknesses) since I am not familiar with the uncertainty/OOD/classification-with-rejection domain.

[1] Jishnu Mukhoti, Viveka Kulharia, Amartya Sanyal, Stuart Golodetz, Philip H. S. Torr, Puneet K. Dokania, Calibrating Deep Neural Networks using Focal Loss, NeurIPS 2020

[2] Rui Huang, Yixuan Li, MOS: Towards Scaling Out-of-distribution Detection for Large Semantic Space, CVPR 2021

Post-rebuttal:

I am not convinced by the rebuttal and choose to be skeptical of the proposed approach.

I think the crux of disagreement lies here:

 -  The trustworthiness scores are indeed dependent on the classifier.

The rebuttal and the paper explicitly say it only depends on the image input which makes it a very different problem formulation departing from previous literature [r1]. I want to stress that the difference is not between access to intermediate layers or the last of the unknown classifier, but that there should be access to the unknown classifier at all. [r1] in contrast does assume access, infact stating that they fix the unknown classifier and make it accessible to the trustworthiness predictor.

In this work as I understand, ideally they should have trained another classifier to predict trustworthiness (of the unknown classifier) given only the soft information of predictions by that unknown classifier on training data. Otherwise it's confusing how a ViT unknown model, ResNet trustworthiness predictor could be used. I hope this clarifies the point I was trying to make.

(Note that [r1] chooses to take the intermediate layers but could just additionally have incorporated the last layer as well, however in this work they cannot assume access to the underlying classifier.)

The defense of miscalibrated classifiers and adversarial attacks seem not convincing to me for two reasons:

- Calibration has been surprisingly accurate and existing methods seem to alleviate the problem to a large degree on both in-distribution and OOD data. It seems like a more straightforward comparison to make if allowed access to the classifier-- which the authors could just say is unfair because that formulation specifically requires access which they cannot assume (which is my original objection).
- Adversarial attacks can be made on the trustworthiness predictor as well. I am pondering why dependence on the classifier makes it different, the trustworthiness predictor could ignore the classifiers outputs entirely and choose to rely more on the image.

**Time Spent Reviewing:**

7

---

> ### Author Response · Authors · 2021-08-10
> **Thank you for the helpful and detailed comments**
>
> We thank the reviewer for the constructive feedback! We are encouraged that the reviewer finds our work "clear and well-presented" with "very significant results and interesting transferability". We are glad that the reviewer recognizes the "reproducibility & nice ablations" in our work. We address the questions/comments below and will incorporate all feedback in the final version.
>
> **DFoG-Q1**: In my opinion, trustworthiness of a classifier’s output depends on the classifier. Consequently, the input to the trustworthiness predictor should have that output of the target classifier passed (softmax probability/logit etc) alongside the image.
>
> **Response**: As elaborated in Section 3.1, we follow the design of TCP [r1]. Specifically, the confidence model (i.e., trustworthiness predictor) only takes images as input. Then, both the output of the trustworthiness predictor and the output of the classifier are used in the loss term related to the trustworthiness.
>
> In contrast, if the output (e.g., softmax probability or logit) of the target classifier is one of the inputs to the trustworthiness predictor, there will be two drawbacks:
>
> 1) As discussed in both TCP [r1] and trust score [r2], a higher confidence score from the classifier does not necessarily imply a higher probability that the classifier is correct. This implies that taking the classifier’s probabilities or logits as input to the trustworthiness predictor may lead to a negative effect on predicting trustworthiness.
>
> 2) Adversarial perturbation has been proven to change the confidence of classifiers dramatically [r3]. The perturbed logit/probability could disturb the trustworthiness prediction.
>
> **DFoG-Q2-part1**:  Past works to the best of my knowledge the trustworthiness is determined by the probability outputs of the target classifiers (a function of the target classifier).
>
> **Response**: As mentioned in the response to Q1, TCP [r1] indicates that a plausible framework to determine the trustworthiness is by the features of images, instead of the probability outputs of the target classifier. In other words, the trustworthiness can’t be reliably determined by the probability output of the target classifiers.
>
> **DFoG-Q2-part2**:  The trustworthiness is typically used for out-of-distribution detection rather than rejecting test samples.
>
> **Response**: We agree with the reviewer and none of the existing works (e.g., TCP[r1], Trust Score [r2], MCP [r4]) or ours is used for rejecting examples. However, we would like to point out that trustworthiness informs whether one can safely use the predictions yielded by AI for making decisions, which is not the goal of out-of-distribution detection. Most existing trustworthiness works (e.g., TCP[r1], Trust Score [r2]) are not used for out-of-the-distribution detection. It is also worth noting that while the existing works evaluate the trustworthiness on the smaller scale and in-distribution images (i.e., training and validation on CIFAR, MNIST, etc), the proposed method is trained with ImageNet images and evaluated on both in-distribution images (i.e., ImageNet images) and out-of-distribution images (stylized and adversarial ImageNet images) to verify its generalizability.
>
> **DFoG-Q2-part3**: The calibrated probability estimates for the same are obtained by using loss functions like cross-entropy (with some calibration) {13} and focal loss [1] -- not training a separate binary classifier with these loss functions as done here, along with assumption of having access to just the image. Recent (parallel) work [2] has benchmarked these methods on large-scale datasets which indicates these scale well to larger datasets. A fair comparison in my opinion should be with (older-- {13},[1]) classifiers using the output of target networks to determine trustworthiness.
>
> **Response**: As mentioned in responses to Q1 and Q2-part1, we choose and follow the design of TCP [r1] over the alternative design due to unreliable classifier’s probabilities [r1, r2] and potential adversarial perturbation [r3] that changes the probability dramatically. Specifically, TCP [r1] and Trust Score [r2] indicate that directly using MCP [r4] (i.e., {13} indicated by the reviewer) from a classifier would be unreliable (e.g., Figure 1 in TCP [r1]). To address this issue, TCP [r1] proposes a framework that consists of a confidence model (i.e., "a separate binary classifier") and a classification model, where both models take images as input.
>
> Calibrating probabilities yielded by a classifier [r5] (i.e., [1] indicated by the reviewer) is not the focus of this work. Instead of training classifiers, we assume that the pre-trained classifiers are well-tuned and publicly available (e.g., ResNet, ViT, etc.). Given such classifiers, we aim to learn corresponding trustworthiness predictors while retaining the pre-trained classifiers. In other words, the classifier calibrated by [r5] could be used as a pre-trained classifier in our proposed framework.
>
> **DFoG-Q3**: No substantial negative societal impact that must've been discussed in my view but the section was missing from the draft.
>
> **Response**: We thank the reviewer for pointing out this, and will add the following section on societal impact.
>
> Learning high-accuracy models is a long-standing goal in machine learning. Nevertheless, due to the complexity of real-world data, there is still a gap between state-of-the-art classification models and a perfect one. Hence, there is a critical need to understand the trustworthiness of classification models, i.e., finding out correct predictions and incorrect predictions, in order to safely and effectively apply the models in real-world tasks. Models with the proposed loss achieve considerably better performance with various trustworthiness metrics. They also show generalizability with both in-distribution and out-of-distribution images at a large scale. In addition, while existing trustworthiness models focus on a high true positive rate (TPR) and tend to view all incorrect predictions to be trustworthy (i.e., TNR close to 0), fasle positives may lead to high consequent cost in some real-world scenarios such as critical applications in medicine (e.g., a false positive leading to unnecessary and invasive tests or treatment such as biopsies or surgery, or harmful side effects in medicine) and security (e.g., counter terrorism). It is thus of great importance for a trustworthiness model to flexibly trade off between TPR and TNR. To this end, the proposed loss allows the underlying distributions of positive and negative examples to be more separable, enabling a more effective trade-off between them.
>
> **References**:
>
> [r1] Charles Corbière, Nicolas Thome, Avner Bar-Hen, Matthieu Cord, and Patrick Pérez. "Addressing Failure Prediction by Learning Model Confidence." NeurIPS 2019.
>
> [r2] Heinrich Jiang, Been Kim, Melody Y. Guan, and Maya R. Gupta. "To Trust Or Not To Trust A Classifier." NeurIPS 2018.
>
> [r3] Ian J. Goodfellow, Jonathon Shlens, and Christian Szegedy. "Explaining and harnessing adversarial examples." arXiv preprint arXiv:1412.6572 (2014).
>
> [r4] Dan Hendrycks and Kevin Gimpel. "A baseline for detecting misclassified and out-of-distribution examples in neural networks." ICLR 2017.
>
> [r5] Jishnu Mukhoti, Viveka Kulharia, Amartya Sanyal, Stuart Golodetz, Philip H. S. Torr, Puneet K. Dokania. "Calibrating Deep Neural Networks using Focal Loss." NeurIPS 2020.

---

> ### Comment · Reviewer_8gNS · 2021-08-24
> **Regarding Weakness 3.1**
>
> I thank the reviewer for a comprehensive review, and wished to clarify a certain aspect of the paper as per my understanding of the methodology. The authors should indeed verify that my understanding here is correct.
>
> Weakness 3.1) "trustworthiness of a classifier’s output depends on the classifier". In my understanding of the SS loss paper, the trustworthiness scores are indeed dependent on the classifier. In my interpretation of the Methodology section, the SS loss paradigm follows the following steps -
> 1. The classifier is trained on the training set using a standard training paradigm,
> 2. the backbone weights of the classifier (everything except the last linear layer) are frozen,
> 3. the trustworthiness classifier is added on top of the frozen backbone and only the weights of the trustworthiness classifier $\theta_{\text{head}}$ are trained using SS loss.
>
> This formulation follows the TCP loss paper [r1] as far as I understand, and seems to be a traditional formulation of trustworthiness established by [r1]. However, my literature survey of this field is not comprehensive, and it is possible there are other formulations as well, that make more sense in certain applications. I evaluated my review based on the improvements of the SS loss within the space of this formulation, and within this scope, the improvements make empirical and theoretical sense. I also don't agree that the trustworthiness predictor should take in the outputs of the classifier, as it is well known that cross-entropy loss softmax outputs are poorly calibrated for in-distribution and OOD settings, and any trustworthiness classifier building only on the softmax outputs would suffer the same issues. It seems that by training a trustworthiness predictor on the high-dimensional representation learned by the classifier instead, you are encoding explicit information about the classifier, without falling prey to issues that the cross-entropy loss is known to result in [r2].
>
> However, it would be interesting to see how much of the trustworthiness performance will change if the trustworthiness classifier is built on top of a trained model that is better calibrated by itself (for example, a model trained using focal loss). However, since the authors used pre-trained backbones (that were presumably trained using CE loss), this seems hard to verify.
>
> [r1] Charles Corbière, Nicolas Thome, Avner Bar-Hen, Matthieu Cord, and Patrick Pérez. "Addressing Failure Prediction by Learning Model Confidence." NeurIPS 2019.
> [r2]  Jishnu Mukhoti, Viveka Kulharia, Amartya Sanyal, Stuart Golodetz, Philip H. S. Torr, Puneet K. Dokania, Calibrating Deep Neural Networks using Focal Loss, NeurIPS 2020.

---

> > ### Author Response · Authors · 2021-08-26
> > **Thanks for the in-depth discussions**
> >
> > We thank the reviewer for valuable input and discussions. We verify that the reviewer’s understanding is correct.
> >
> > First, "the trustworthiness scores are indeed dependent on the classifier" is true. Both the TCP loss [r1] and the proposed SS loss take three inputs into account including the classifier output. Specifically, the inputs include the outputs $f_{\theta}(x)$ of the trustworthiness predictor, the outputs $f_{\theta'}^{(cls)}(x)$ of the classifier, and the ground-truth label $y$, to compute the loss w.r.t. trustworthiness.
> >
> > Second, "This formulation follows the TCP loss paper [r1] as far as I understand, and seems to be a traditional formulation of trustworthiness established by [r1]" is also correct. We follow the formulation of [r1], where (1) both the trustworthiness predictor and classifier take the images as input, and (2) both the TCP loss and the proposed SS loss take the outputs of the trustworthiness predictor, the outputs of the classification model, and the ground-truth label as input (detailed above).
> >
> > The steps summarized by the reviewer are correct for both the TCP loss and the proposed SS loss using the experimental protocol in [r1] on small-scale datasets. With this setting, we showed in Table A2 that the proposed SS loss improves performance. In addition, instead of targeting small-scale datasets (e.g., MNIST, CIFAR) in [r1], our work targets large-scale datasets (e.g., ImageNet). To this end, we made a few slight modifications on the training scheme details, without changing the TCP formulation and flow, to adapt to larger data with flexibility and generality, namely (1) leveraging publicly available classifiers pre-trained on ImageNet, (2) removing the restrictions that backbones of the classifier and trustworthiness predictor have to be identical, and (3) allowing all the parameters of our trustworthiness predictor to be trainable to accommodate the much-increased data volume and diversity in ImageNet. We also showed that with the same revised training scheme, the SS loss performs better than the other losses (i.e., TCP loss, cross entropy loss, and focal loss). Overall, the proposed SS loss is generic that shows advantages for both small-scale and large-scale settings, confirming the reviewer’s acknowledgment that "​​the improvements of the SS loss within the space of this formulation, and within this scope, the improvements make empirical and theoretical sense."
> >
> > Last but not least, regarding the comment "However, it would be interesting to see how much of the trustworthiness performance will change if the trustworthiness classifier is built on top of a trained model that is better calibrated by itself (for example, a model trained using focal loss)." we thank the reviewer for this interesting idea, and for the thoughtfulness to mention "However, since the authors used pre-trained backbones (that were presumably trained using CE loss), this seems hard to verify": indeed, training a classifier on large-scale datasets involves considerable engineering tricks and efforts to achieve the accuracy as high as those pre-trained by Google, FAIR, OpenAI, etc. Our work for now focuses on effectively training a trustworthiness predictor instead of training the classifier, and we will pursue the suggested analysis in the extension of our work.

---

> ### Author Response · Authors · 2021-09-07
> **Response to "post-rebuttal"**
>
> We thank the reviewer for the post-rebuttal comments. We address the points below.
>
> **Comment 1**: I think the crux of disagreement lies here: The trustworthiness scores are indeed dependent on the classifier.
>
> The rebuttal and the paper explicitly say it only depends on the image input which makes it a very different problem formulation departing from previous literature [r1]. I want to stress that the difference is not between access to intermediate layers or the last of the unknown classifier, but that there should be access to the unknown classifier at all. [r1] in contrast does assume access, in fact stating that they fix the unknown classifier and make it accessible to the trustworthiness predictor.
>
> In this work as I understand, ideally they should have trained another classifier to predict trustworthiness (of the unknown classifier) given only the soft information of predictions by that unknown classifier on training data. Otherwise it's confusing how a ViT unknown model, ResNet trustworthiness predictor could be used. I hope this clarifies the point I was trying to make.
>
> **Response**: To clarify some misunderstandings that the reviewer may have, we would like to re-emphasize that: (1) We follow the design of TCP [r1], without changing its formulations or flows. (2) Using the output of the classifier as input has two drawbacks as shown in [r1], [r2], and our work, and thus both [r1] and our work adopt an alternative design. To avoid redundancy, please refer to detailed elaborations of both points with response to DFoG-Q1 ([link](https://openreview.net/forum?id=cBWFSWwjBSC&noteId=TF6-YLMzIyt)), and the discussions with Reviewer m14X (["After rebuttal comments"](https://openreview.net/forum?id=cBWFSWwjBSC&noteId=RYve4SDZKk))
> , and Reviewer 8gNS ([link](https://openreview.net/forum?id=cBWFSWwjBSC&noteId=dDj6UDVNXm)), which advocated the approach of our work.
>
> **Comment 2**: Note that [r1] chooses to take the intermediate layers but could just additionally have incorporated the last layer as well, however in this work they cannot assume access to the underlying classifier.
>
> **Response**: [r1] does not "just additionally have incorporated the last layer as well" for a reason. Specifically, [r1] intentionally not to use the last layer to address drawbacks of directly using MCP confidences, e.g., in page 2 of [r1], it says "MCP confidence values for erroneous and correct predictions overlap. It is worth mentioning that this problem comes from the fact that MCP leads by design to high confidence values, even for erroneous ones, since the largest softmax output is used". It echoes our responses to the reviewer's question (see the first link above).
>
> **Comment 3**: Calibration has been surprisingly accurate and existing methods seem to alleviate the problem to a large degree on both in-distribution and OOD data. It seems like a more straightforward comparison to make if allowed access to the classifier-- which the authors could just say is unfair because that formulation specifically requires access which they cannot assume (which is my original objection).
>
> **Response**: The focus of this work is to study how to train a trustworthiness predictor on large-scale datasets, instead of training a classifier for classification. Therefore, we use publicly available pre-trained classifiers (e.g., ResNet and ViT) that are widely-used in machine learning.
>
> Though not the focus of the work, we may want to experiment with the suggested idea as a future effort: calibration [1] recommended by the reviewer does not provide experimental results on ImageNet, nor pre-trained classifiers, and pre-training a high-accuracy classifier on ImageNet takes a considerable amount of time, which may not be feasible to finish in a week.
>
> **Comment 4**: Adversarial attacks can be made on the trustworthiness predictor as well. I am pondering why dependence on the classifier makes it different, the trustworthiness predictor could ignore the classifier's outputs entirely and choose to rely more on the image.
>
> **Response**: As mentioned in [r1], [r2] and demonstrated in detail in our work, the adversarial perturbation can change the confidence of classifiers dramatically. With the approach in the proposed work, the negative effect of the adversarial perturbation on trustworthiness prediction is reduced as fooling a classifier is much easier than fooling both the classifier and the trustworthiness predictor. Specifically, the architecture of the trustworthiness predictor could be different from the one of the classifier, and the task of predicting trustworthiness is different from the task of classification. As shown in Table 2 of the manuscript, adversarial examples [33] lead to a drop to 6% in accuracy in classification, but the trustworthiness predictor yields TPR (87%) and TNR (24%) with the adversarial examples.
>
> We agree with the reviewer that "the trustworthiness predictor could ignore the classifier's outputs entirely and choose to rely more on the image", which as we wrote in the previous responses, is one reason we chose this design over its counterpart using the classifier's output as input.
>
> **References**:
>
> [1] Jishnu Mukhoti, Viveka Kulharia, Amartya Sanyal, Stuart Golodetz, Philip H. S. Torr, Puneet K. Dokania, Calibrating Deep Neural Networks using Focal Loss, NeurIPS 2020
>
> [r1] Charles Corbière, Nicolas Thome, Avner Bar-Hen, Matthieu Cord, and Patrick Pérez. "Addressing Failure Prediction by Learning Model Confidence." NeurIPS 2019.
>
> [r2] Heinrich Jiang, Been Kim, Melody Y. Guan, and Maya R. Gupta. "To Trust Or Not To Trust A Classifier." NeurIPS 2018.
>
> [33] Cassidy Laidlaw and Soheil Feizi. "Functional adversarial attacks." NeurIPS 2019.

---

### Decision · Program_Chairs · 2021-09-27

**Decision:**

Accept (Poster)

**Comment:**

This paper develops a novel loss function for predicting the "trustworthiness" of a classifier.  This involves training a separate classifier, on existing predictions and corresponding outcomes, that makes a binary prediction about whether a predicted class can be trusted.  The authors show that existing loss functions do not generalize well and instead propose a novel steep slope loss.  The authors present theoretical generalization bounds and empirical evidence on a variety of non-IID validation sets that their method outperforms existing losses.  Overall the reviewers thought that the paper was well written, the contributions significant and the empirical evidence convincing.
 One criticism was regarding the amount of access required by the method to the classifier of interest (there was significant debate about this and seems to have been resolved to an acceptable level).   Two reviewers voted high confidence accepts (7, 8) and two low confidence borderline scores (5, 6).  Thus the recommendation is to accept the paper.   Please try to incorporate feedback from the reviewers (e.g. clarification about access to the classifier) in the camera ready version.